# CROSS-WINDOW SELF-TRAINING VIA CONTEXT VARIATIONS FROM SPARSELY-LABELED TIME SERIES

## ABSTRACT

A real-world time series is often sparsely labeled due to the expensive annotation cost. Recently, self-training methods have been applied to a dataset with few labels to infer the labels of unlabeled augmented instances. Accelerating this trend for time-series data, fully taking advantage of its sequential nature, we propose a novel data augmentation approach called *context-additive augmentation*, which allows a target instance to be augmented easily by adding preceding and succeeding instances to form an augmented instance. Unlike the existing augmentation techniques which may alter the target instance by directly perturbing its features, it preserves a target instance as is but still gives various augmented instances with varying contexts. Additionally, we propose a *cross-window self-training* framework based on the context-additive augmentation. The framework first augments target instances by applying context-varying windows over a given time series. Then, the framework derives reliability-based cross-window labels and uses them to maintain consistency among augmented instances across the windows. Extensive experiments using real datasets show that the framework outperforms the existing state-of-the-art self-training methods.

## 1 INTRODUCTION

A time series is a collection of consecutive data points, often annotated with temporally coherent timestamp labels[R4:, and this work deals with a model aiming to classify every timestamp in a time series correctly.] However, due to the length of and complexity in a time series, labeling every timestamp in the time series requires prohibitively high cost, and therefore, in reality a lot of time series are only sparsely labeled (Moltisanti et al., 2019; Ma et al., 2020; Deldari et al., 2021; Shin et al., 2022). In this regard, self-training is used as a promising way to train a model from sparse labels, by leveraging the model's output to infer new labels for unlabeled data points (Laine & Aila, 2017; Rizve et al., 2021). Recent state-of-the-art self-training methods, mostly developed for image data, necessitate domain-specific data augmentation (Sohn et al., 2020; Zhang et al., 2021; Kim & Lee, 2022).

Such conventional data augmentation generates multiple different instances from a target instance [R2,R4: (i.e., an instance for pseudo-labeling)] by way of data perturbation. If data instances are independent of one another as in image data, there is no other way than to perturb the target instance itself. In contrast, using the sequential nature of time series, where data instances (segments or data points) are temporally correlated, it is feasible to generate multiple different instances from a target instance without perturbing it but by adding its surrounding sequence (i.e., context). [R2,R4: As shown in Figure 1(a), given a target instance sampled from a time series, contexts of varying lengths are added to the preceding and succeeding positions of the target instance to generate different pairs of "augmented" instances.] We call this type of data augmentation the *context-additive augmentation*.

The key property of context-additive augmentation is to achieve the effect of data augmentation *without* perturbing a target instance. Being free of data perturbation brings several benefits. First, consistency between augmented instances can be enforced more reliably because a target instance itself is exactly the same among its augmented instances. Second, a sufficient number of augmented instances can be easily obtained by *context variations*. Third, it is computationally inexpensive, only requiring the retrieval of a sub-sequence from a time series. Moreover, context-additive augmentation can be used together with conventional data augmentation such as jittering and scaling. Thus,

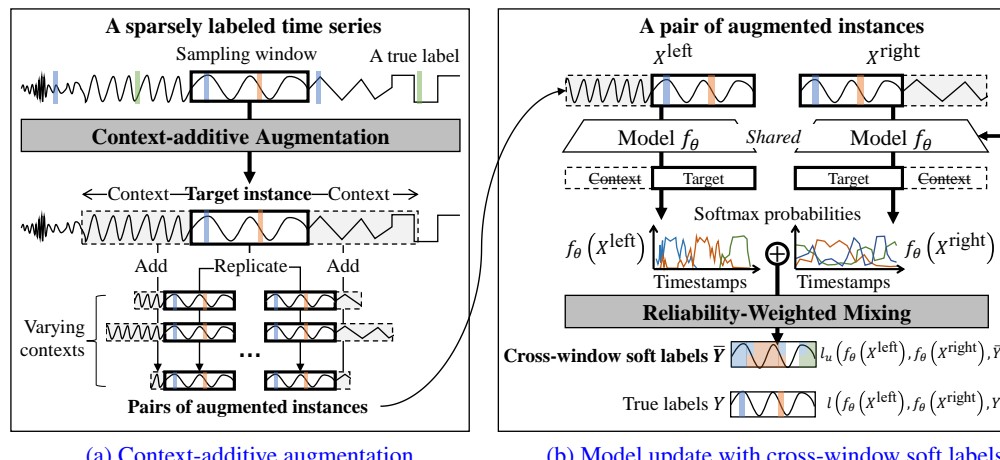

Figure 1: [R1,R2,R3,R4:]Illustration of CrossMatch.

the novel concept of context-additive augmentation opens a new direction of data augmentation for sequential data, i.e., time series.

Despite its time-series-savvy concept and big benefits, applying context-additive augmentation for self-training is challenging. First, it requires determining the proper number and range of context variations based on the trade-off between the expected performance improvement and training cost. Note that varying the context length in augmented instances incurs different complexity for a downstream task; intuitively, compared with the conventional data augmentation, a short context gives weak augmentation and a long context gives strong augmentation [R4:] considering the intensity of perturbation. Second, a new consistency regularization method is needed to fully take advantage of the benefit of context-additive augmentation, which does not perturb a target instance unlike the conventional data augmentation.

In order to address the aforementioned challenges, we propose a novel self-training approach, called *CrossMatch*, for time series. In existing self-training methods such as MixMatch (Berthelot et al., 2019) and FixMatch (Sohn et al., 2020), an artificial label is created as a form of a *hard* label; because the model's outputs for augmented instances could be biased by the perturbation of the target instance, the most confident label is chosen by averaging and sharpening in MixMatch and weak augmentation and thresholding in FixMatch. In CrossMatch, on the other hand, due to its *target-preserving* property, the model's outputs with the contexts of the same length are considered equally meaningful, and therefore, the model's output from an augmented instance (i.e., window) is *crossed* to other augmented instances in the form of a *soft* label. [R2:] As shown in Figure 1(b), a pair of augmented instances generated from a target instance is fed to a model to get the two softmax outputs of the target instance. Then, a single set of *cross-window* soft labels is derived and enforced to each output for consistency regularization. The same procedure repeats using diverse pairs of augmented instances.

In summary, for time-series self-training, CrossMatch conducts context-additive augmentation with varying contexts and consistency regularization among augmented instances using cross-window soft labels. Especially for the first aforementioned challenge, we empirically analyze the impact of context variations on classification accuracy in Section 4.2. Through extensive evaluation using three sparsely-labeled time-series datasets, despite its simplicity CrossMatch is shown to achieve higher classification accuracy than the existing state-of-the-art methods, significantly outperforming the FixMatch style methods with jittering and scaling up to 23p%.

## 2 RELATED WORK

### 2.1 DATA AUGMENTATION

Data augmentation perturbs given data instances to generate diverse and sufficient data instances to prevent overfitting (Shorten & Khoshgoftaar, 2019). The techniques used in data augmentation

usually manipulate features and can be classified into two categories: (1) inner-augmentation that changes the features within a data instance and (2) inter-augmentation that exploits features across multiple data instances. Rotation, flipping, scaling, cutout, and random erasing are the examples of inner-augmentation (DeVries & Taylor, 2017; Cubuk et al., 2019). Mix-up, cut-mix, and copy-paste are the representatives of inter-augmentation that mixes two full or partial images (Zhang et al., 2018; Yun et al., 2019; Ghiasi et al., 2021). However, these studies have not considered context-additive augmentation because they deal with independent data instances such as images.

Time-series augmentation techniques have been devised in recent literature (Wen et al., 2021). They include jittering, scaling, window warping, window cropping, and the Fourier transform specialized in the time and frequency domains, and belong to the inner-augmentation category (Um et al., 2017; Eldele et al., 2021; Yue et al., 2022; Chen et al., 2022). The inter-augmentation category is considered to be ineffective because temporal patterns could be lost after mixing two time-series segments (Iwana & Uchida, 2021). These studies also consider a set of already segmented time series as candidate augmentation targets and assume that the instances in each segment have the same labels. Thus, they are not directly applicable to a *single* continuous time series with *sparse* labels, which is a more practical and challenging setting. Moreover, all of them perturb the target instances, following the common trend of existing data augmentation.

## 2.2 SEMI-SUPERVISED LEARNING BASED ON AUGMENTATIONS

Semi-supervised learning (SSL) trains a model with both labeled and unlabeled data instances (Van Engelen & Hoos, 2020). Unlabeled data instances are harnessed by two popular approaches: (1) self-supervised learning which is mostly based on contrastive learning or pretext tasks (Chen et al., 2020; Grill et al., 2020; Singh et al., 2021) and (2) self-training which produces artificial pseudo-labels for unlabeled data instances from model predictions (Lee, 2013; Chen et al., 2018b;a; Xie et al., 2020; Pham et al., 2021). We focus on self-training owing to its simplicity and effectiveness demonstrated in recent studies (Yang et al., 2021).

State-of-the-art self-training methods force consistency in model predictions from multiple augmentations of a data instance. MixMatch (Berthelot et al., 2019) averages out the predictions from multiple augmentations and sharpens the averaged prediction to reduce the entropy in the pseudo-label. ReMixMatch (Berthelot et al., 2020) generates a sharpened pseudo-label from a weak augmentation and matches it against the predictions from multiple strong augmentations. FixMatch (Sohn et al., 2020) generates a one-hot pseudo-label by choosing a single class above a *fixed* confidence threshold. FlexMatch (Zhang et al., 2021) is a variation of FixMatch, which uses a *dynamic* confidence threshold to adapt to different learning speeds among different classes. Propagation regularizer (Kim & Lee, 2022) also reduces confidence in incorrect predictions to make FixMatch robust to a more sparse label setting. However, all these methods heavily rely on domain-specific augmentation and lack for consideration of time series.

There are several time-series semi-supervised learning methods in the literature, but most of them are based on self-supervised learning. In their settings, a model is first pre-trained with time-series self-supervision and then fine-tuned with initial labels. The examples of time-series self-supervisions are (1) pretext tasks such as forecasting and temporal relation prediction (Jawed et al., 2020; Fan et al., 2021), (2) contrastive learning with the aforementioned time-series augmentation techniques (Singh et al., 2021; Xiao et al., 2022), and (3) clustering results (Singhania et al., 2022). These methods target an already-segmented time series, which cannot deal with continuous time series with sparse labels (Ma et al., 2021; Goschenhofer et al., 2021; Xu et al., 2022).

## 3 CROSSMATCH: CROSS-WINDOW TIME-SERIES SELF-TRAINING

### 3.1 PRELIMINARIES AND PROBLEM SETTING

Table 1 summarizes basic notations used in this paper.

**Dataset and Model:** Let $\mathcal{D} = \mathcal{X} \times \mathcal{Y} = \{(\boldsymbol{x}_t, y_t) \mid t \in \mathcal{T}\}$ be a time series, where $\mathcal{T}$ is an index set of timestamps, $\boldsymbol{x}_t \in \mathbb{R}^d$ is a $d$-dimensional data point at timestamp $t$, and $y_t$ is a corresponding class label if $\boldsymbol{x}_t$ is labeled or null otherwise. Let $\mathcal{T}_L$ be the index set of *labeled* timestamps and $\mathcal{T}_U$ be the index set of *unlabeled* timestamps, where $\mathcal{T}_L \cup \mathcal{T}_U = \mathcal{T}$ and $|\mathcal{T}_L| \ll |\mathcal{T}_U|$. Here, $\mathcal{T}_L$ is sparse, that

is, its members are few and scattered. In this work, multiple consecutive timestamps, referred to as a *segment instance*, is usually processed in a batch. An instance $X = \{\boldsymbol{x}_t \mid t \in [m-w : m+w)\}$ is a set of consecutive $2w$ data points (timestamps) with $d$ features sliced from $\mathcal{D}$, where $[m-w : m+w)$ represents an integer interval from $m-w$ through $m+w-1$. Likewise, $Y = \{y_t \mid t \in [m-w : m+w)\}$ is a set of the corresponding class labels sliced from $\mathcal{D}$, where $y_t \in \{1, \ldots, K\}$ and $K$ is the number of classes. A model $f_\theta$ predicts the sequential softmax probabilities of $X$, i.e., $f_\theta(X) \in [0, 1]^{2w \times K}$. The classification loss for training the model given $X$ and $Y$ is formulated as

$$\ell(X, Y) = \frac{1}{2w} \sum_t \mathbf{1}_{y_t \neq \texttt{null}} H(f_\theta(X)_{t,:}, y_t), \qquad (1)$$

where $H(\cdot, \cdot)$ is sparse categorical cross-entropy and $\cdot_{t,:}$ means indexing at timestamp $t$.

**Pseudo-labeling:** For each instance $X$, using the maximum softmax probabilities conditioned on a confidence threshold $\tau$, a pseudo-label $\hat{y}_t$ at each timestamp $t \in [m-w : m+w)$ is derived by

$$\hat{y}_t = \begin{cases} \arg\max_{k \in \{1, \ldots, K\}} f_\theta(X)_{t,k} & \text{if } f_\theta(X)_{t,k} > \tau \\ \texttt{null} & \text{otherwise.} \end{cases} \qquad (2)$$

A set $\hat{Y} = \{\hat{y}_t \mid t \in [m-w : m+w)\}$ of the pseudo-labels for $X$ is constructed by Equation (2). Then, the classification loss for an instance $X$ and its set $\hat{Y}$ of the pseudo-labels obtained is formulated as $\ell(X, \hat{Y})$ with Equation (1). **[R2,R4:]** Note that we do not store those pseudo-labels but discard them after the model update from $\ell(X, \hat{Y})$ whenever a target instance is sampled for pseudo-labeling.

Table 1: A summary of notations.

| | |
|---|---|
| $\boldsymbol{x}_t \in \mathbb{R}^d$ | a data point at timestamp $t$ |
| $X$ | an instance, a set of data points |
| $\dot{X}$ | a target instance |
| $X'$ | an augmented instance |
| $t$ | a timestamp |
| $w$ | half of the length of $X'$ |
| $m$ | the middle timestamp of $X$ |
| $o$ | half of the length of $\dot{X}$ |
| $c$ | half of the length of context, $w - o$ |

**Consistency regularization:** Recent self-training methods force consistency between the model outputs of augmentations as the class information is preserved across augmentations. For example, FixMatch matches the pseudo-label from a weak augmentation against the prediction from a strong augmentation (Sohn et al., 2020). That is, $\ell(X, \hat{Y}) = \frac{1}{2w} \sum_t \mathbf{1}_{\max_k f_\theta(\alpha(X))_{t,k} > \tau} H(f_\theta(\mathcal{A}(X))_t, \hat{y}_t)$, where $\hat{y}_t = \arg\max_k f_\theta(\alpha(X))_{t,k}$, and $\alpha$ and $\mathcal{A}$ represent weak and strong augmentations. **[R2,R4:]** The goal of consistency regularization is to offer informative supervision to update the model by diverse augmentations and reliable pseudo-labels. In the example of FixMatch, pseudo-labels from weak augmentations are reliable due to weak perturbation, and strong augmentations become diverse due to strong perturbation.

### 3.2 CONTEXT-ADDITIVE AUGMENTATION

Given a *target instance* $\dot{X} = \{\boldsymbol{x}_t \mid t \in [m-o : m+o)\}$, we add a context of total length $c$ to its surroundings to generate an augmented instance $X' = \{\boldsymbol{x}_t \mid t \in [m-o-c_l : m+o+c_r)\}$. There are numerous design choices for generating $X'$ where the context lengths $c_l$ and $c_r$ control the degree of perturbation and the allocation of a context around $\dot{X}$ determines the multiplicity of augmentation. Increasing the multiplicity between augmented instances helps derive more informative pseudo-labels, but this high multiplicity also increases the redundancy between similar augmented instances and causes inefficient computations.

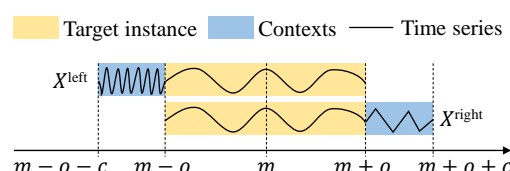

Figure 2: Illustration of a target instance and its context.

In this work, we follow a simple heuristic for the allocation of the context **[R1,R2,R4:]** for effective consistency reguluarization. A context of a given length $c$ is added on the left and right sides of the target instance, and thus two augmented instances, $X^{\text{left}} = \{x_t \mid t \in [m-o-c : m+o)\}$ and $X^{\text{right}} = \{x_t \mid t \in [m-o : m+o+c)\}$, are generated as shown in Figure 2. **[R1,R2,R3,R4:]** The underlying reason is that forcing consistency between the model outputs from the augmented instances with more difference gives more strong supervision to the model (Wang et al., 2022). This heuristic is practical and reasonable, as it not only maximizes the difference between the two

augmented instances but also increases efficiency in further model inference due to small number of augmented instances. Each augmented instance is fed to a model, and pseudo-labels are generated only from the target instance (i.e., the overlap between the two augmented instances) by Equation (2). As a result, two pseudo-label sets $\hat{Y}^{\text{left}}$ and $\hat{Y}^{\text{right}}$ for the left and and right augmented instances, respectively, are obtained.

The value of $o$, which is half of the length of a target instance, needs to be carefully chosen for each data set. Too large $o$ may include semantically irrelevant data points, while too small $o$ may not give enough temporal information for prediction. Either way leads to incorrect pseudo-labels and ultimately degrades the performance. Thus, we set $2o$, the length of a target instance, as a value lower than the mean length of a label-coherent segment, which could be known in advance or estimated by a given set $\mathcal{T}_L$ of labeled timestamps.

More important factor is the length of context $c$, which directly affect the information quality of consistency regularization. If $c$ is too short, then two augmented instances become too similar so that the model barely changes after matching the outputs of augmented instances (Wang et al., 2022). On the other hand, if $c$ is too long, the outputs diverge so that the matching would weld the representation of instances from different class. We empirically study this trade-off in Section 4.4. To prevent the potential divergence in self-training, we devise reliability-weighted mixing of pseudo-labels generated from two augmented instances.

### 3.3 Reliability-Weighted Mixing

Though all pseudo-labels generated for a target instance satisfy the confidence threshold in Equation (2), we treat them differently based on the reliability of a pseudo-label. Our rationale is that the pseudo-label becomes more reliable if (1) the model $f_\theta$ receives a larger number of data points on the left and right sides of the pseudo-label and (2) the number of data points is balanced between the two sides so that the prediction is not biased to the preceding or succeeding interval. We design a reliability function that follows our rationale and

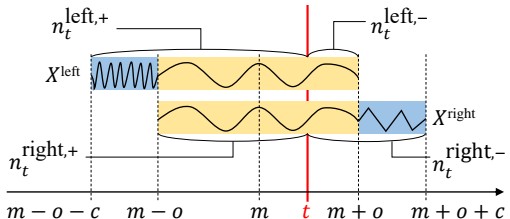

Figure 3: Visualization of Equation (3).

compute reliability of each pseudo-label. Using the reliability as a weight, we mix the pseudo-labels into a single *cross-window label*, which will be matched against two softmax probabilities from both of augmentations.

Let's consider two pseudo-labels $\hat{y}_t^{\text{left}}$ and $\hat{y}_t^{\text{right}}$ generated from $f_\theta(X^{\text{left}})_t$ and $f_\theta(X^{\text{right}})_t$ shown in Figure 3. For each timestamp $t \in [m-o : m+o]$ of a target instance, the length $n_t^{loc,+}$ of the left side and the length $n_t^{loc,-}$ of the right side along each augmented instance are easily calculated by

$$n_t^{loc,+} = \begin{cases} t-m+o+c & \text{if } loc = \text{left} \\ t-m+o & \text{if } loc = \text{right} \end{cases} \quad \text{and} \quad n_t^{loc,-} = \begin{cases} m+o-t & \text{if } loc = \text{left} \\ m+o+c-t & \text{if } loc = \text{right}, \end{cases} \quad (3)$$

where $loc$ indicates the location of context addition—either left or right.

Per our design rationale, the reliability score becomes higher as $n_t^{loc,+}$ and $n_t^{loc,-}$ are larger and $n_t^{loc,+}$ and $n_t^{loc,-}$ are more similar with each other. This requirement can be achieved by exploiting a bell-shaped function,

$$r(p) = \sqrt{2p - p^2} + \sqrt{1 - p^2}, \quad (4)$$

where $0 \le p \le 1$. Here, $p$ is a normalization of $n_t^{loc,+}$ with respect to the full length of an augmented instance, i.e., $p^{loc}(t) = n^{loc,+}/(2o + c)$. That is, using Equation (4), we obtain two reliability scores $r(p^{\text{left}}(t))$ and $r(p^{\text{right}}(t))$ for each pseudo-label from the two augmented instances $X_{left}$ and $X_{right}$. Due to the bell shape in Figure 4, a timestamp with a sufficiently long side length on both sides has an adequately high reliability in an augmented instance. If a side length on either side is too long (i.e.,

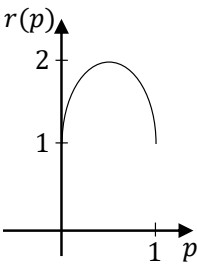

Figure 4: Equation (3).

biased to one side), then the side length on the other side is too short and consequently the reliability score is too low. The reliability score is maximized at the center of a given augmented instance.

Last, in order to treat each pseudo-label differently based on the reliability scores on the two sides, $r(p^{\text{left}}(t))$ and $r(p^{\text{right}}(t))$, the weight of each pseudo-label is assigned as $w_t^{loc} = r(p^{loc}(t))/(r(p^{\text{left}}(t)) + r(p^{\text{right}}(t)))$ by normalizing the two reliability scores. The final *cross-window soft label* for the target instances in the left and right instances is

$$\bar{y}_t = \mathcal{N}\big(w_t^{\text{left}} \cdot \texttt{onehot}(\hat{y}_t^{\text{left}}) + w_t^{\text{right}} \cdot \texttt{onehot}(\hat{y}_t^{right})\big), \tag{5}$$

where the function $\mathcal{N}$ normalizes the input to make the sum of the softmax probabilities as 1 and the $\texttt{onehot}$ function converts a scalar to one-hot encoded vector. In other words, reliability-weighted mixing generates a soft label by adding and normalizing the two pseudo-labels. Using the cross-window label set and two target instance outputs, we compute classification loss as follows:

$$\ell_u(X^{\text{left}}, X^{\text{right}}, \bar{Y}) = \frac{1}{4o} \sum_t \mathbf{1}_{\bar{y}_t \neq \texttt{null}}\big(h(f_\theta(X^{\text{left}})_{t,:}, \bar{y}_t) + h(f_\theta(X^{\text{right}})_{t,:}, \bar{y}_t)\big), \tag{6}$$

where $h$ is soft cross-entropy and $\bar{Y} = \{\bar{y}_t | t \in [m-o : m+o)\}$. By matching a single cross-window soft label to both augmented instances, we can reduce the variance of the label to be predicted. In addition, self-traning with soft labeling is more effective than with hard labeling since soft labeling can be robust to erroneous pseudo-labels when at least one pseudo-label is correct (Müller et al., 2019).

### 3.4 OVERALL SELF-TRAINING PROCEDURE USING CROSSMATCH

**R1,R2,R3,R4:** Figure 1 summarizes the overall procedure of CrossMatch using context-additive augmentation and reliability-weighted mixing. Given a time series, CrossMatch first takes a target instance sampled from the time series as an input and generates a pair of augmented instances with context-additive augmentation (see Figure 1(a)). Then, a model $f_\theta$ infers the pair of augmented instances $X^{\text{left}}$ and $X^{\text{right}}$ to produce two independent softmax probabilities. The pseudo-labels respectively generated from the two softmax probabilities are averaged ($\bigoplus$) to give cross-window soft labels $\bar{Y}$ with reliability-weighted mixing. Finally, the cross-entropy losses are computed from the cross-window soft labels $\bar{Y}$ as well as the true labels $Y$ and used to update the model $f_\theta$ (see Figure 1(b)). These steps are repeated with a randomly sampled target instance for $I$ iterations. CrossMatch in Appendix A details each step.

## 4 EVALUATION

### 4.1 EXPERIMENT SETTING

**Datasets:** We use three widely-used benchmark datasets summarized in Table 2. mHealth is an action recognition dataset recorded with wearable sensors, such as 3D accelerometers, 3D gyroscopes, 3D magnetometers, and electrodes, whose sampling frequency is

Table 2: Datasets and configurations.

| | $|\mathcal{T}|$ | Length | #class | $d$ | $o$ | $c$ | $z$ | $I$ |
|---|---|---|---|---|---|---|---|---|
| HAPT | 408K | 967 | 6 | 6 | 512 | 256 | 0.1% | 25K |
| mHealth | 343K | 2933 | 12 | 23 | 384 | 256 | 1.0% | 50K |
| Opportunity | 190K | 109 | 17 | 113 | 512 | 64 | 1.0% | 30K |

50Hz (Banos et al., 2014). HAPT is also a sensor time-series dataset tracking human movements in a laboratory sampled with the frequency of 50Hz (Anguita et al., 2013). Opportunity is a collection of sensor recordings at 100Hz capturing daily natural human activities with wearable, object, and ambient sensors (Roggen et al., 2010). **R2,R4:** For each originally fully-labeled dataset, we randomly sample the same number of timestamps for each class and drop the labels from the rest of the timestamps to generate a *sparsely-labeled* time series, leaving multiple labeled time spans located randomly. The sampled timestamps become a labeled timestamp set $\mathcal{T}_L$ for the given ratio of labeled timestamps $z = |\mathcal{T}_L|/|\mathcal{T}|$. Table 2 summarizes the statistics of datasets and the data-specific parameters, specifying the number of timestamps, the average length of a segment with a single class, the number of classes, the dimension of a data point, the half length of a target instance $o$, the context length $c$, the default ratio of labeled timestamps $z$, and the maximum number of iterations $I$.

**Implementation Details:** We use a popular multi-stage temporal convolutional network named MS-TCN (Farha & Gall, 2019), which is applicable to our problem setting as it gives softmax prob-

Table 3: Timestamp accuracy and F1@25 score averaged over the last 20 iterations with adjusting the label ratio $z$ ($1\times$, $5\times$, and $10\times$ of a default value). The best values are marked in bold.

| Method | Ratio | FixMatch | | FlexMatch | | PropReg | | CrossMatch | |
|---|---|---|---|---|---|---|---|---|---|
| | | TS Accuracy | F1@25 | TS Accuracy | F1@25 | TS Accuracy | F1@25 | TS Accuracy | F1@25 |
| HAPT | $1\times$ | $0.78 \pm 0.01$ | $0.55 \pm 0.04$ | $0.78 \pm 0.01$ | $0.54 \pm 0.04$ | $0.79 \pm 0.01$ | $0.52 \pm 0.04$ | $\mathbf{0.84 \pm 0.01}$ | $\mathbf{0.75 \pm 0.02}$ |
| | $5\times$ | $0.85 \pm 0.01$ | $0.51 \pm 0.04$ | $0.84 \pm 0.01$ | $0.50 \pm 0.04$ | $0.82 \pm 0.01$ | $0.49 \pm 0.04$ | $\mathbf{0.88 \pm 0.01}$ | $\mathbf{0.70 \pm 0.02}$ |
| | $10\times$ | $\mathbf{0.96 \pm 0.00}$ | $0.87 \pm 0.01$ | $\mathbf{0.96 \pm 0.00}$ | $\mathbf{0.90 \pm 0.01}$ | $\mathbf{0.96 \pm 0.00}$ | $0.89 \pm 0.01$ | $0.95 \pm 0.01$ | $0.84 \pm 0.02$ |
| mHealth | $1\times$ | $0.77 \pm 0.01$ | $0.23 \pm 0.01$ | $0.77 \pm 0.01$ | $0.13 \pm 0.01$ | $0.81 \pm 0.01$ | $0.43 \pm 0.03$ | $\mathbf{0.85 \pm 0.01}$ | $\mathbf{0.45 \pm 0.02}$ |
| | $5\times$ | $0.91 \pm 0.01$ | $0.65 \pm 0.02$ | $0.91 \pm 0.01$ | $0.62 \pm 0.03$ | $0.91 \pm 0.01$ | $0.67 \pm 0.03$ | $\mathbf{0.94 \pm 0.01}$ | $\mathbf{0.74 \pm 0.03}$ |
| | $10\times$ | $0.91 \pm 0.00$ | $0.70 \pm 0.02$ | $0.90 \pm 0.00$ | $0.66 \pm 0.02$ | $0.90 \pm 0.01$ | $0.70 \pm 0.02$ | $\mathbf{0.95 \pm 0.01}$ | $\mathbf{0.84 \pm 0.03}$ |
| Opportunity | $1\times$ | $0.61 \pm 0.04$ | $0.65 \pm 0.05$ | $0.59 \pm 0.03$ | $0.63 \pm 0.05$ | $0.63 \pm 0.03$ | $0.65 \pm 0.05$ | $\mathbf{0.67 \pm 0.02}$ | $\mathbf{0.73 \pm 0.04}$ |
| | $5\times$ | $0.73 \pm 0.03$ | $0.73 \pm 0.04$ | $0.72 \pm 0.03$ | $0.75 \pm 0.05$ | $0.73 \pm 0.03$ | $0.74 \pm 0.05$ | $\mathbf{0.78 \pm 0.02}$ | $\mathbf{0.82 \pm 0.03}$ |
| | $10\times$ | $0.75 \pm 0.03$ | $0.75 \pm 0.04$ | $0.73 \pm 0.03$ | $0.77 \pm 0.04$ | $0.75 \pm 0.03$ | $0.76 \pm 0.04$ | $\mathbf{0.83 \pm 0.02}$ | $\mathbf{0.86 \pm 0.02}$ |

abilities for each timestamp in an input instance $X$. We follow the same hyperparameter and configuration in the original MS-TCN, except the learning rate and an optimizer adjusted for a self-training environment with sparse labels. Please refer to Appendix B for more details.

For CrossMatch, we set the confidence threshold $\tau$ to $0.95$ and the weight of the unlabeled loss $\lambda$ to $1$. The model is first trained without self-training, i.e., only using the labeled batches (Algorithm 1 Line 6–7). We start to update a model with pseudo-labels after the number of pseudo-labels in each class for a batch is balanced. Formally, this condition is satisfied when the entropy of the numbers of pseudo-labels per class is above $0.99$ for the last $100$ iterations; it is enforced to prevent early confirmation bias in self-training (Kim & Lee, 2022). If a data point $x_t$ in an instance $X$ has a true label (i.e., $t \in \mathcal{T}_L$), CrossMatch uses the true label instead of the generated pseudo-label.

**Evaluation Metrics:** [R1:]*We measure* timestamp accuracy *and* segmental F1 score *with five-fold cross validation and report the average value with standard deviation of five runs. For sequential classifiers such as MS-TCN, timestamp accuracy (denoted as TS accuracy) and segmental F1 score (denoted as F1@25) measure the performance of classification at each timestamp and segment respectively (Li et al., 2021; Kumar et al., 2022). To evaluate pseudo-labeling performance, we report* pseudo-label F1 score *(denoted as PLF) as well. We define each metric in detail in Appendix C.*

**Compared Self-Training Methods:** We compare CrossMatch with three state-of-the-art self-training methods: FixMatch (Sohn et al., 2020), FlexMatch (Zhang et al., 2021), and PropReg (Kim & Lee, 2022). As discussed in Section 2, these methods require inner-instance augmentation for consistency regularization. We use two popular time-series augmentations: jittering and scaling, where weak augmentation $\alpha(X) = \text{jittering}(X)$ and strong augmentation $\mathcal{A}(X) = \text{jittering}(\text{scaling}(X))$ (Um et al., 2017). Throughout the experiments, we use the same hyperparameters for all methods except the instance length due to the context-additive augmentation. As an additional evaluation for fair comparison, we also compare CrossMatch with the variants of the compared methods which are modified in support of our context-additive augmentation.

### 4.2 COMPARISON WITH STATE-OF-THE-ART SELF-TRAINING METHODS

**Overall Comparison:** Table 3 shows timestamp accuracy and F1@25 on three datasets with varying label ratios; each value is obtained by averaging over the last 20 iterations for reliable results. Compared with other methods, CrossMatch achieves the best classification performance [R1:] *with a statistical significance of $0.05$ using independent (unpaired) t-test for all datasets except HAPT $10\times$.* This is mainly because consistency regularization using context-additive augmentation is more informative than inner-instance augmentation used in other self-training methods. In particular, Cross-Match exhibits much better performance than others especially when the initial label ratio is low. For instance, with only $0.1\%$ of the labeled timestamps in HAPT, CrossMatch outperforms FixMatch by 20p%, FlexMatch by 21p%, and PropReg by 23p% in F1@25 (see the first row in HAPT data of Table 3). Therefore, the performance dominance of CrossMatch indicates the context-additive augmentation with reliability-weighted mixing indeed helps the model select more reliable pseudo-labels even when softmax probabilities fluctuate due to label scarcity.

**Training Curve Analysis:** Figure 5 shows the training curves of classification and PLF over the entire training iteration. Please refer to Appendix D for the same results with standard deviation and [R2,R3,R4:]*other metrics related to pseudo-labeling.* CrossMatch shows [R3:] *much higher performance than the other methods with respect to timestamp accuracy and F1@25 even in the early*

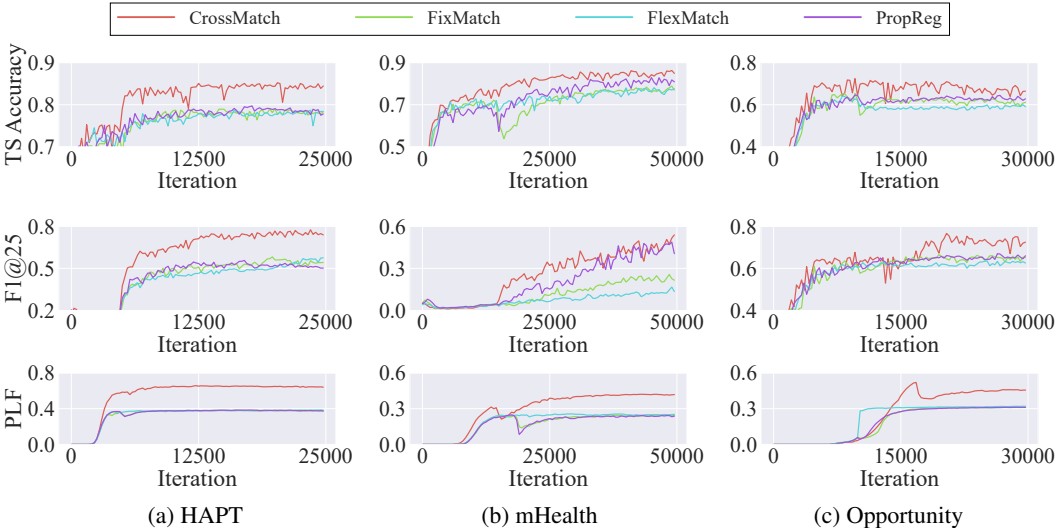

Figure 5: Training curve of the classification performance (the first two rows) and [R1,R2:] pseudo-labeling performance (the last row) over self-training iterations in Algorithm 1.

stage of training, reaching the highest performance in most cases (see the first two rows in Figure 5). This is attributed to its robustness in reliability-weighted mixing in the early stage of self-training and its credibility in enforcing consistency between two augmented instances from context-additive augmentation. For example, as shown in mHealth of Figure 5(b), other methods suffer from the low-quality pseudo-labels at the iteration of around $17,000$ when the warm-up period ends, thereby showing a temporary drop in timestamp accuracy. Although the accuracy recovers gradually, the final accuracy is far behind the accuracy CrossMatch achieves.

The effectiveness of CrossMatch is also supported by [R2,R3,R4:] its PLF computed from precision and recall (see the last row in Figure 5 and Figure 7). Precision is the ratio of the number of correctly predicted timestamps to the number of pseudo-labeled timestamps, while recall is the ratio of the number of correctly predicted timestamps to the length of a target instance (refer to Appendix C for more details). We averaged PLF over the target instances in a batch. CrossMatch reaches the highest PLF continuously in most cases due to the reliability of our cross-window labels.

## 4.3 Extensions with Inner- and Inter-instance Augmentation

We investigate another possible extensions: (1) CrossMatch to combine the two inner-instance augmentation (+IA) of jittering and scaling and (2) the variant of existing self-training methods to combine our proposed context-additive augmentation (+CA). For the former, we perform jittering and scaling before applying the context-additive augmentation. For the latter, we modify the existing methods in support of the context-additive augmentation. The two augmented instances (the left and right instances in Figure 2) are respectively treated as weakly and strongly augmented instances for existing methods; the instance with a higher reliability score than the other becomes the weakly augmented instance.

Table 4: Classification accuracy of the compared methods with context-additive augmentation (CA) and CrossMatch with inner-instance augmentation (IA).

| Dataset | HAPT | | mHealth | |
| --- | --- | --- | --- | --- |
| | TS Accuracy | F1@25 | TS Accuracy | F1@25 |
| CrossMatch | $0.84 \pm 0.01$ | $0.75 \pm 0.02$ | $0.85 \pm 0.01$ | $0.45 \pm 0.02$ |
| CrossMatch+IA | $0.88 \pm 0.01$ | $0.77 \pm 0.02$ | $0.89 \pm 0.01$ | $0.67 \pm 0.03$ |
| FixMatch | $0.78 \pm 0.01$ | $0.55 \pm 0.04$ | $0.77 \pm 0.01$ | $0.23 \pm 0.01$ |
| FixMatch+CA | $0.62 \pm 0.02$ | $0.39 \pm 0.03$ | $0.76 \pm 0.04$ | $0.35 \pm 0.04$ |
| FlexMatch | $0.78 \pm 0.01$ | $0.54 \pm 0.04$ | $0.77 \pm 0.01$ | $0.13 \pm 0.01$ |
| FlexMatch+CA | $0.64 \pm 0.02$ | $0.49 \pm 0.03$ | $0.82 \pm 0.01$ | $0.31 \pm 0.03$ |
| PropReg | $0.79 \pm 0.01$ | $0.52 \pm 0.04$ | $0.81 \pm 0.01$ | $0.43 \pm 0.03$ |
| PropReg+CA | $0.68 \pm 0.01$ | $0.45 \pm 0.02$ | $0.77 \pm 0.03$ | $0.30 \pm 0.03$ |

Table 4 summarizes the performance of possible extensions for HAPT and mHealth datasets with their default label ratios of $0.1\%$ and $1.0\%$. First, CrossMatch with jittering and scaling shows additional performance gain which is about $4p\%$ in timestamp accuracy and $2–22p\%$ in F1@25 for the two datasets. These improvements demonstrate that jittering and scaling can be applied together with context-additive augmentation without performance drop under the proposed self-

Table 5: Timestamp accuracy and F1@25 averaged over the last 20 iterations with varying context lengths using CrossMatch. The best values are marked in bold.

| $c$ | HAPT | | mHealth | | Opportunity | |
|---|---|---|---|---|---|---|
| | TS Accuracy | F1@25 | TS Accuracy | F1@25 | TS Accuracy | F1@25 |
| $0.25\times$ | $0.78 \pm 0.01$ | $0.67 \pm 0.01$ | $0.69 \pm 0.01$ | $0.29 \pm 0.02$ | $0.65 \pm 0.03$ | $0.72 \pm 0.04$ |
| $0.50\times$ | $0.79 \pm 0.02$ | $0.65 \pm 0.03$ | $0.79 \pm 0.02$ | $0.37 \pm 0.02$ | $\mathbf{0.69 \pm 0.03}$ | $\mathbf{0.73 \pm 0.04}$ |
| $1.00\times$ | $0.81 \pm 0.01$ | $0.70 \pm 0.02$ | $\mathbf{0.85 \pm 0.01}$ | $\mathbf{0.45 \pm 0.02}$ | $0.67 \pm 0.02$ | $0.73 \pm 0.04$ |
| $2.00\times$ | $\mathbf{0.84 \pm 0.01}$ | $\mathbf{0.78 \pm 0.01}$ | $0.82 \pm 0.01$ | $0.38 \pm 0.04$ | $0.66 \pm 0.02$ | $0.71 \pm 0.03$ |
| $4.00\times$ | $0.84 \pm 0.01$ | $0.76 \pm 0.02$ | $0.77 \pm 0.02$ | $0.28 \pm 0.03$ | $0.55 \pm 0.02$ | $0.56 \pm 0.04$ |

Table 6: [R1,R2,R3:]Timestamp accuracy and F1@25 averaged over the last 20 iterations with the variations of context-additive augmentation (CA). The context length for fixed-context CA is set as the best value observed in Table 5. The best results are marked in bold.

| Variations | Left-only CA | | Right-only CA | | Fixed-context CA | | CrossMatch | |
|---|---|---|---|---|---|---|---|---|
| | TS Accuracy | F1@25 | TS Accuracy | F1@25 | TS Accuracy | F1@25 | TS Accuracy | F1@25 |
| HAPT | $0.72 \pm 0.01$ | $0.65 \pm 0.03$ | $0.73 \pm 0.02$ | $0.67 \pm 0.02$ | $0.78 \pm 0.01$ | $0.65 \pm 0.02$ | $\mathbf{0.84 \pm 0.01}$ | $\mathbf{0.75 \pm 0.02}$ |
| mHealth | $0.77 \pm 0.02$ | $0.41 \pm 0.04$ | $0.76 \pm 0.02$ | $0.43 \pm 0.05$ | $0.73 \pm 0.02$ | $0.22 \pm 0.02$ | $\mathbf{0.85 \pm 0.01}$ | $\mathbf{0.45 \pm 0.02}$ |

training framework. However, in general, the simple extensions of existing methods for context-additive augmentation rather suffer from a significant performance drop. Therefore, our method is robust to the addition of existing inner-instance augmentation.

## 4.4 ANALYSIS OF VARYING CONTEXT LENGTHS

Table 5 summarizes the timestamp accuracy and F1@25 score of CrossMatch by adjusting the default context length $c$ (in Table 2) from $0.25\times$ to $4.0\times$. If the context length is too large, augmented instances bear too much perturbation to generate high-quality cross-window labels. On the other hand, if the context length is too small, informative consistency regularization between two augmented instances becomes trivial since they show high similarity. We found out that the optimal context length is highly correlated with the average segment length of each dataset. For instance, the best timestamp accuracy of Opportunity data is achieved with a relatively smaller context length (i.e., $64 \times 0.5 = 32$) than that of mHealth data (i.e., $256 \times 1.0 = 256$) because Opportunity data has much shorter mean segment length; as can be seen in Table 2, Opportunity and mHealth exhibit the shortest and the longest mean segment length among the three datasets. Therefore, the best timestamp accuracy and F1@25 score of each dataset is achieved with different context lengths.

[R1,R2,R3:] Table 6 shows the classification performance of CrossMatch with the variations of context-additive augmentation (CA). The context is added to only either side of a target instance (i.e., *left-only CA* or *right-only CA*), and the length of the context is fixed rather than varying (i.e., *fixed-context CA*). We set the fixed length to the best value found in Table 5. These variations weaken the diversity of augmented instances, which degrade the effect of consistency regularization, so all of them result in lower classification performance than CrossMatch.

## 5 CONCLUSION

In this paper, we propose a novel time-series self-training method CrossMatch equipped with context-additive augmentation. It adds a context instance to a target instance on its left and right sides within a window to generate two augmented instances with different contexts. To reduce variance in pseudo-labeling, CrossMatch mixes the two sets of pseudo-labels obtained from the two augmented instances with inferred reliability scores. Our extensive experiments demonstrate that CrossMatch achieves considerably higher classification accuracy than other state-of-the-art self-training methods—by up to 23p% even when only 0.1% of timestamps are labeled. CrossMatch introduces a new direction of data augmentation for sequential data and has the potential to be applied to a variety of time-series applications. For further work, we plan theoretical analysis of the optimal context length and performance maximization of self-training using context-additive augmentation.

## CODE OF ETHICS

In this paper, we propose a time-series self-training framework with context-additive augmentation. Our algorithm improves model performance despite sparsity of labels in time-series data, and this improvement can reduce the cost of label annotations. In experiments we use popular time-series datasets open to the public and available from the websites at the links provided in the references. Each human subject in the datasets is encoded as a random number for anonymization.

## REPRODUCIBILITY

We elaborate on the details of our algorithm and dataset setting in the main text for reproduction. Section 4.1 and Appendix B describe the dataset setting for sparsely-labeled time series and explain hyperparameters used in the training and configuration. Algorithm 1 summarizes our framework as a pseudo-code for better understanding. Finally, our work can be reproduced using the guidelines and the source codes in `https://www.dropbox.com/sh/j2n2hrfbze39ags/AADCRye_a-4pIomnc9OWtG2qa?dl=0`.

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

# A    OVERALL TRAINING ALGORITHM

---

**Algorithm 1** Time-series self-training with CrossMatch

---

**Input:** A time series with initial labels $\mathcal{D}$, labeled timestamp set $\mathcal{T}_L$, unlabeled timestamp set $\mathcal{T}_U$,
   labeled batch size $B_l$, unlabeled batch size $B_u$, a model $f_\theta$, confidence threshold $\tau$,
   loss weight $\lambda$, learning rate $\eta$, half the length of target instance $o$, context length $c$,
   max number of iterations $I$.
**Output:** Final model $f_\theta$.
1:  **for** each iteration up to $I$ **do**
2:     $\mathcal{T}_l \leftarrow$ Sample $B_l$ timestamps from $\mathcal{T}_L$; $\mathcal{T}_u \leftarrow$ Sample $B_u$ timestamps from $\mathcal{T}_U$;
3:     $c' \leftarrow$ Sample a context length from Uniform$(2, c)$;
4:     $w \leftarrow o + c'$;
5:     /** Loss computation for labeled batch **/
6:     $\mathcal{X}_l \leftarrow \{\mathcal{X}_{[m-w:m+w]}, m \in \mathcal{T}_l\}$; $\mathcal{Y}_l \leftarrow \{\mathcal{Y}_{[m-w:m+w]}, t \in \mathcal{T}_l\}$;
7:     $\mathcal{L}_l = \frac{1}{B_l} \sum_{X \in \mathcal{X}_l, Y \in \mathcal{Y}_l} \ell(X, Y)$;                    // See Equation (1)
8:     /** Loss computation for unlabeled batch **/
9:     $\mathcal{X}_u \leftarrow \{(\mathcal{X}_{[m-w:m+o]}, \mathcal{X}_{[m-o:m+w]}), m \in \mathcal{T}_u\}$;                    // See Section 3.2
10:    $\hat{\mathcal{Y}}_u \leftarrow$ PSEUDOLABELING$(\mathcal{X}_u, f_\theta, \tau)$;                    // See Equation (2)
11:    $\bar{\mathcal{Y}} \leftarrow$ RELIABILITYWEIGHTING$(\hat{\mathcal{Y}}_u)$;                    // See Section 3.3
12:    $\mathcal{L}_u = \frac{1}{B_u} \sum_{(X^{\text{left}}, X^{\text{right}}) \in \mathcal{X}_u, \bar{Y} \in \bar{\mathcal{Y}}} \ell_u(X^{\text{left}}, X^{\text{right}}, \bar{Y})$;                    // See Equation (6)
13:    $\mathcal{L} \leftarrow \mathcal{L}_l + \lambda \mathcal{L}_u$; $\theta \leftarrow \theta - \eta \nabla_\theta \mathcal{L}$;
14: **return** $f_\theta$;

---

Algorithm 1 describes how CrossMatch works in time-series self-training. For each iteration, there are two steps for batch initialization: center timestamp sampling from $\mathcal{T}_L$ and $\mathcal{T}_U$ (Line 2) and context length sampling from a uniform distribution (Line 3). Note that we assign $w$ as $o + c'$ after the sampling (Line 4). From labeled middle timestamps, instances are sliced from $\mathcal{X}$ with length $2w$ to construct a labeled batch, $\mathcal{X}_l$ and $\mathcal{Y}_l$, where $\mathcal{X}_{[m-w:m+w]} = \{\boldsymbol{x}_t \mid t \in [m-w : m+w)\}$ and $\mathcal{Y}_{[m-w:m+w]} = \{y_t \mid t \in [m-w : m+w)\}$ (Line 6). The classification loss for a labeled batch $\mathcal{L}_l$ is computed and averaged over the batch (Line 7). From unlabeled middle timestamps, instances are sliced with context-additive augmentation that generates a pair of instances whose length is $o + w$ (Line 9). The target instance of each instance in an augmented unlabeled batch $\mathcal{X}_u$ is then pseudo-labeled using a confidence threshold $\tau$ (Line 10). CrossMatch softens the pseudo-labels with reliability weighting across two instances with different contexts, transforming two pseudo-labels into a single cross-window label shared across the instances (Line 11). The classification loss with cross-window labels is computed for each pair of augmented instances and is averaged over the batch (Line 12). Finally, the losses for labeled and unlabeled batches are then integrated into a single loss $\mathcal{L}$ with a hyperparameter $\lambda$, and the model $f_\theta$ is updated using its gradient (Line 13).

# B    DETAILS OF TRAINING THE CLASSIFIER

R2: As described in implementation details of 4.1, we use MS-TCN as backbone sequential classifier (Farha & Gall, 2019). It can classify each data point in a segment instance $X$, generating sequential softmax probabilities at each timestamp. MS-TCN has four stages, and each stage is composed of eleven dilated convolution layers and a softmax output layer. The first stage takes a subsequence of the whole time series and outputs softmax probability distribution at each timestamp. After the first stage, every stage is fed with softmax probabilities and then outputs another softmax probabilities. For all datasets, we use the same training hyperparameters and classifier as listed in Table 7. We set the labeled batch size as 4 and the unlabeled batch size as 8, use SGD optimizer with momentum and Nesterov method. The initial learning rate is 0.005 and is scheduled with a cosine decay function. The letter $i$ in Scheduling denotes the current iteration number during training, and the letter $I$ denotes the total number of iterations.

Table 7: Training hyperparameters.

| Stage | Layer | $B_L$ | $B_U$ | Optimizer | Momentum | Nesterov | $\eta$ | Scheduling |
|-------|-------|-------|-------|-----------|----------|----------|--------|------------|
| 4 | 11 | 4 | 8 | SGD | 0.9 | True | 0.005 | $cos(\frac{7\pi i}{I})$ |

## C  DETAILS IN EVALUATION METRICS

[R1]: Timestamp accuracy is the ratio of the number of timestamps with correctly predicted labels to the total number of timestamps in a times series, computed as follows.

$$\text{TS accuracy} = \frac{1}{|\mathcal{T}_{\text{test}}|} \sum_{t \in \mathcal{T}_{\text{test}}} \mathbf{1}_{y_t = \hat{y}_t}$$

Segmental F1 score is a performance measure for judging whether a classifier outputs *correct and coherent* labels for consecutive timestamps. Segmental precision and recall are first computed by counting the number of correct matches between predicted segments set $\hat{\mathbb{Y}}$ and true segments set $\mathbb{Y}$ with Jaccard similarity threshold (here, we set $0.25$) as follows.

$$\text{Precision} = \frac{1}{|\hat{\mathbb{Y}}|} \sum_{\hat{Y} \in \hat{\mathbb{Y}}} \sum_{Y \in \mathbb{Y}} \mathbf{1}_{\text{Jaccard}(\hat{Y}, Y) > 0.25} \qquad \text{Recall} = \frac{1}{|\mathbb{Y}|} \sum_{\hat{Y} \in \hat{\mathbb{Y}}} \sum_{Y \in \mathbb{Y}} \mathbf{1}_{\text{Jaccard}(\hat{Y}, Y) > 0.25}$$

Finally, Segmental F1 score is computed as $F1@25 = \frac{2*\text{precision}*\text{recall}}{\text{precision}+\text{recall}}$.

[R2]: At each iteration, we measure pseudo-label precision and pseudo-label recall from each times-tamp of target instances in a batch, as follows:

$$\text{PL Precision} = \frac{\text{the number of correct PLs}}{\text{the number of PLs}} \qquad \text{PL Recall} = \frac{\text{the number of correct PLs}}{\text{the number of timestamps}}$$

For *every* timestamp, we check the existence of a pseudo-label and the class of the true label to see if the class of a pseudo-label is correct to count the numerators and denominators in the above formula. When there is no pseudo-label at any timestamp, the denominator of precision becomes 0 and the precision can diverge. So, we define the precision with no pseudo-label as 0. Finally, pseudo-label F1 score is $\text{PLF} = \frac{2*\text{PL precision}*\text{PL recall}}{\text{PL precision}+\text{PL recall}}$. After a few iterations, the entire data is expected to be used by pseudo-labeling the sampled target instances.

## D  DETAILED ILLUSTRATIONS

Figure 6 is a detailed version of Figure 5, with standard deviation error bars added. [R2,R3,R4]Figure 7 shows the pseudo-labeling metrics such as precision, recall, and the total number of pseudo-labels. Figure 8 shows the accuracy convergence trend when context length is varied as a factor or multiple of the context length $c$, illustrating accuracy curve during the training iterations.

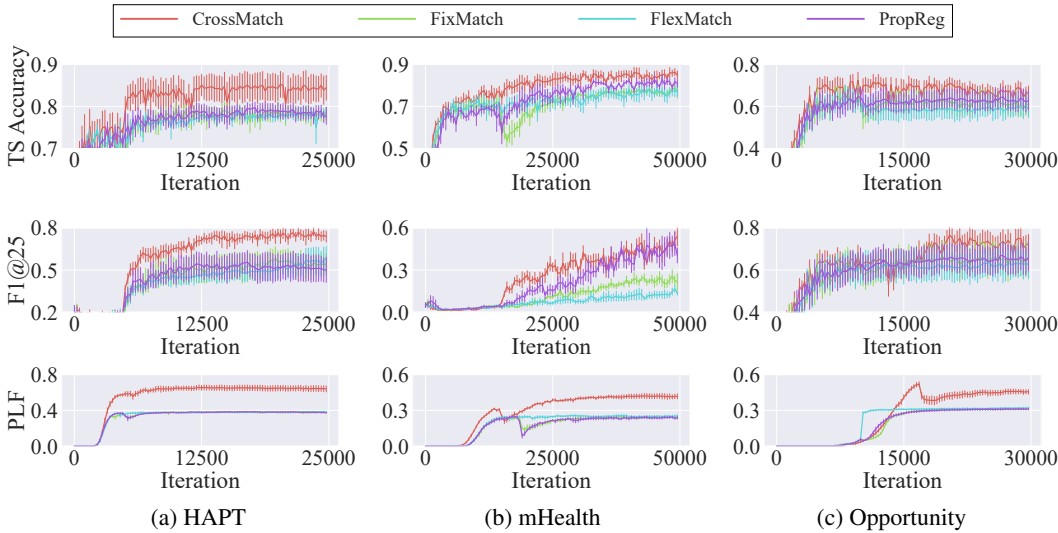

Figure 6: Training curve of the classification performance (the first two rows) and pseudo-labeling performance (the last row) over self-training iterations in Algorithm 1. Error bars represent the standard deviation of five runs.

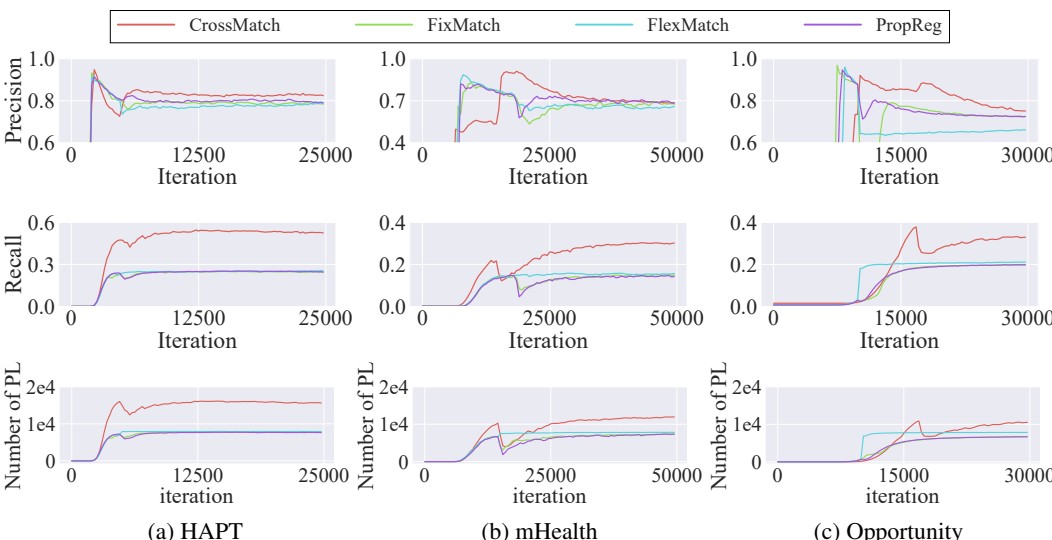

Figure 7: Precision, recall, and the number of pseudo-labels of CrossMatch compared with those of FixMatch, FlexMatch, and PropReg.

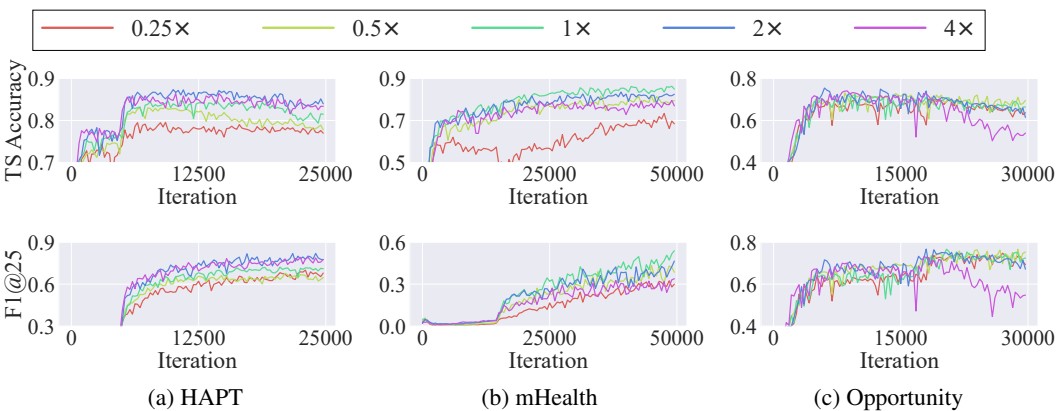

Figure 8: Classification accuracy of CrossMatch for varying context length $x \times c$.

