# OpenReview forum: "Cross-Window Self-Training via Context Variations from Sparsely-Labeled Time Series"
_ICLR.cc/2023/Conference — Submitted to ICLR 2023_

### Official Review · Reviewer_Rri3 · 2022-10-17

**Confidence:** 3
**Correctness:** 2
**Technical Novelty And Significance:** 3
**Empirical Novelty And Significance:** 2
**Recommendation:** 5

**Clarity, Quality, Novelty And Reproducibility:**

The paper would benefit from clarification throughout to make the model more intuitive.

The work proposes an interesting idea that would benefit from further comparisons with label propagation methods and clarification of the experimental settings. For instance, the studied datasets present multiple individuals, but it is said that "all included time series are concatenated in chronological order to form a single continuous time series", what does it mean in this context?
It would also be beneficial to display the number of pseudo labels used for each method (if possible having a fixed number across methodologies to compare the quality of the data added, which may be confounded by the number of points added). Formalisations of the assumptions made on the time series and the label missingness would be key to understanding the limitations of the proposed methodology.

The code is provided for reproducibility.



**Strength And Weaknesses:**

The paper explores an important problem as time series often present sparse labelling.

However, the clarity of the paper weakens its impact. The introduction contains all necessary elements but would benefit from clarifications and examples to give the intuition behind the proposed heuristic. "Target instance", "by adding surrounding sequences", and "weak and strong augementaions" are terms that need to be more clearly introduced. The specific problem should also be introduced earlier, the paper seems to focus on temporal prediction for which each time step needs to be classified. At the end of the introduction, it is still not clear what challenges the proposed method addresses and what the motivation for this approach is.

**Summary Of The Paper:**

This paper introduces the concept of context additive augmentation for self-supervised training for sparsely labelled time series. The idea is to propagate the labels on surrounding timesteps, and then used the pseudo labels for self-training. This work explores multiple real-world datasets to compare performance against other state-of-the-art approaches.

**Summary Of The Review:**

The paper proposes an interesting heuristic that could be strengthened by clarifications.

---

> ### Author Response · Authors · 2022-11-16
> **Response to Reviewer Rri3**
>
> We deeply appreciate the reviewer's contructive comments on our manuscript.
>
> **Q1. "Target instance", "by adding surrounding sequences", and "weak and strong augmentations" are terms that need to be more clearly introduced.**
>
> > Target instance.
>
> A target instance is an instance we would like to do pseudo-labeling, where its only few timestamps might have been labeled. We clarified this meaning in Section 1 and Figure 1(a).
>
> > Adding surrounding sequences.
>
> Context-additive augmentation adds surrounding sequences whose timestamps are before the start timestamp of a target instance and after the end timestamp of a target instance.
>
> > weak/strong augmentation.
>
> Strong and weak augmentation is determined by the intensity of data perturbation. Let us have two augmentation, one is jittering $A_1(x)=x+\epsilon$ where $\epsilon$ is a noise term and another is scaling $A_2(x)=\gamma x$. As the sequential use of two augmentations $A_2(A_1(x))$ perturbs the data $x$ more than a single augmentation $A_1(x)$ does, we can say $A_2(A_1(x))$ is a strong augmentation and $A_1(x)$ is a weak augmentation. These terms are widely used in recent consistency regularization works such as [Sohn et al., 2021]. For clarity, we added more explanation on the terms you mentioned in Section 1.
>
>
> **Q2.The specific problem should also be introduced earlier, the paper seems to focus on temporal prediction for which each time step needs to be classified.**
>
> As you suggested, we introduce the specific problem in Section 1 as follows: "The goal of time-series classification is to predict every timestamp in a time series correctly."
>
>
> **Q3. At the end of the introduction, it is still not clear what challenges the proposed method addresses and what the motivation for this approach is.**
>
> In this paper, we tried to design effective context-additive augmentation and generate reliable labels for consistency regularization when self-training from sparsely-labeled time series. Context-additive augmentation generates augmented instances with maximal difference in a practical way and generate a shared label called as cross-window label considering context-additive augmentation. We have revised Introduction and Figure 1 according to the reviewers' comments, and thus hope that the draft reads more smoothly to you.
>
>
> **Q4. The work proposes an interesting idea that would benefit from further comparisons with label propagation methods.**
>
>
> Thanks for the suggestion. We think it is a great future research topic that we would like to pursue. In our problem setting, we generate pseudo-labels directly from the model and update the model instantaneously, different from label propagation paradigm where propagated labels are accumulated to update the model occationally [Iscen, et al., 2019]. We will compare CrossMatch with other label propagation methods in near future.
>
> **Q5. For instance, the studied datasets present multiple individuals, but it is said that "all included time series are concatenated in chronological order to form a single continuous time series", what does it mean in this context? Formalizations of the assumptions made on the time series and the label missingness would be key to understanding the limitations of the proposed methodology.**
>
> Thank you for pointing out our mistake in writing. We did not concatenate all the instances. For each fully-labeled dataset, we randomly sample the equal number of timestamps for each class and drop the labels at the rest of timestamps to generate *sparsely-labeled* time series. The sampled timestamps becomes a labeled timestamp set $\mathcal{T}_L$ given the ratio of labeled timestamps $z=|\mathcal{T}_L|/|\mathcal{T}|$, varying from 0.1% to 10% for each experiment. We corrected our manuscript accordingly in Section 4.1.
>
>
> [Sohn et al., 2020] Fixmatch: Simplifying semi-supervised learning with consistency and confidence. In NeurIPS, 2020
>
> [Iscen, et al., 2019]  Label propagation for deep semi-supervised learning. In CVPR. 2019.

---

> > ### Author Response · Authors · 2022-11-16
> > **Additional Response to Reviewer Rri3**
> >
> > **Q6. It would also be beneficial to display the number of pseudo labels used for each method (if possible having a fixed number across methodologies to compare the quality of the data added, which may be confounded by the number of points added).**
> >
> > Since all compared algorithms generate pseudo-labels in an online manner while a model is being updated, the number of pseudo-labels cannot be fixed in advance. We believe that the quality of pseudo-labels is more important than their quantity since wrong pseudo-labels may make the model fall into a negative feedback loop and degrade the performance rapidly. So, we measured precision, recall, and F1 scores of pseudo-labels along with classification performance over iterations. As suggested, we also measured the number of pseudo labels in each algorithm, which shows a similar trend to the recall score over iterations. Please refer to Figure 5 (F1 score) and Figure 7 (precision, recall, and the number). In Figure 7, except in the HAPT dataset, the numbers of pseudo-labels are not that different across the methods, and thus the higher accuracy of CrossMatch is not simply confounded by the number of pseudo-labels generated.

---

> > > ### Comment · Reviewer_Rri3 · 2022-12-05
> > > **Clarification**
> > >
> > > Thank you for including this, I do agree that the quality of labels is more important than quantity. However, it seems hard to disentangle the two in the proposed figure in which the proposed methodology always leads to more pseudo-labels. Therefore, it is hard to know if the algorithm provides a real edge or if the choice of the parameters leads to more labelled data and therefore the observed performance difference.

---

> > > > ### Author Response · Authors · 2022-12-05
> > > > **About pseudo-label quality and experiment setting**
> > > >
> > > > First of all, we apply the same confidence threshold $\tau=0.95$ (Equation 2) throughout whole experiments and the same number of initial labels are used in every method for a fair comparison. Note that the labels of y-axis in Figure 7 are pseudo-label precision and pseudo-label recall(refer to Appendix C for the definitions). Sorry for the confusion in those labels.
> > > >
> > > > From the definitions, high PL precision and high PL recall mean that a method generates correct PL without missing most of them, thereby improving the quality and quantity of pseudo-labels [Lokhande et al, 2020]. If a classifier generates numerous correct pseudo-labels at a training iteration, those pseudo-labels can train the classifier to become more confident and accurate, generating higher prediction probability for a correct class at the next iteration. This positive feedback loop explains the superiority of CrossMatch in pseudo-label precision, pseudo-label recall, and the number of pseudo-labels.
> > > >
> > > > We tried our best to make the comparison fair enough by selecting the same criteria for generating pseudo-labels. In addition, the number of pseudo-labels was not configured manually throughout experiments, but they are just the result of each method.
> > > >
> > > > [Lokhande, et al., 2020] Generating accurate pseudo-labels in semi-supervised learning and avoiding overconfident predictions via Hermite polynomial activations. In CVPR. 2020.

---

> > > > > ### Comment · Reviewer_Rri3 · 2022-12-05
> > > > > **Clarification**
> > > > >
> > > > > Thank you, the text would benefit from clarifications on this point and analysis of these plots (I thought the recall and precision were associated with the task, not the pseudo-labelling.) Why do we observe a systematic drop? How did you choose the number of iterations?

---

> > > > > > ### Author Response · Authors · 2022-12-06
> > > > > > **About systematic drop and iterations**
> > > > > >
> > > > > > > Clarification of precision and recall
> > > > > >
> > > > > > Thanks for the suggesetion. We will clarify that recall and precision in Figure 7 are associated with pseudo-labeling, not the task.
> > > > > >
> > > > > > > Systematic drop
> > > > > >
> > > > > > Only a few labels are used for training the classifier at the early stage of self-training. however, when each method starts pseudo-labeling (after the warm-up phase), many pseudo-labels are additionally used to help train for better generalization, which causes a big change in the model's parameters possibly overfitted to the few initial labels. At the point, a significant drop occurs.
> > > > > >
> > > > > > > Choosing the number of iterations
> > > > > >
> > > > > > Iteration is a smaller unit for the number of repetitions compared with Epoch (i.e., $\text{1 Epoch}=\frac{|\mathcal{T}|}{2o*B}\text{Iteration}$). We ran every dataset for about 25--50 epochs, similar to FixMatch where the number of epochs for supervised learning is used [Sohn et al., 2020].
> > > > > >
> > > > > > [Sohn et al., 2020] Fixmatch: Simplifying semi-supervised learning with consistency and confidence. In NeurIPS, 2020.

---

> > ### Comment · Reviewer_Rri3 · 2022-12-05
> > **Response**
> >
> > Thank you for your answers and for the overall improvement of the paper.
> > I would like to maintain my rating as the paper's clarity still weakens the work, and the new results question if the edge is the result of more labelled data. The work would benefit from a fairer comparison with state-of-the-art approaches.

---

### Official Review · Reviewer_Cx6c · 2022-10-23

**Confidence:** 4
**Clarity, Quality, Novelty And Reproducibility:** Good
**Correctness:** 3
**Technical Novelty And Significance:** 3
**Empirical Novelty And Significance:** 3
**Recommendation:** 6

**Strength And Weaknesses:**

Strength:
1. The idea of reliability-weighted mixing is interesting. This can help readers design other soft-mixing methods.
2. The results of the experiments show that the proposed method achieves SOTA performance.

Weakness:
1. One might don't agree with the rationale in Section 3.3 and the curve r(p) should be. In my opinion, the reliability value of the middle point of the augmented instance is maximum is not proper. In my opinion, the overlap ( m-o to m+o) should have the largest reliability. If the rationale you proposed holds, assuming that c < 2o, the m - c/2 -th position on X_left will have much larger reliability than the same position but on X_right. Please explain the intuition of your rationale.
2. It is confused with the augmentation method you proposed. Assuming that the target instance is X[m-o, m+o], the generated instances are X[m-o-c, m+o] and X[m-o, m+o+c]. In my option, this is equal to the case that the target is X[m-o-c, m+o+c] and cut the target into X[m-o-c, m+o] and X[m-o, m+o+c]. So I think the novelty is limited.
3. The statements in Section 3 sometimes are not clear enough. The details can be referred to as the minor weaknesses below. Please modify the statements in order to make the paper easily understood by the readers.

Minors:
1. In Section 3.2, the authors say that the heuristic "not only maximizes the difference between the two augmented instances but also minimizes the overlap between them". However, I am not sure what is the difference between "maximizing the difference" and "minimizing the overlap". I think the authors just expressed the same meaning in two ways. If there is some difference, please explain it in detail.
2. The purpose of Figure 5 is unclear. Since the accuracy has been shown in the tables, and it is difficult to see that CrossMatch converges most quickly. Even if it converges quickly, the speed of convergence is likely related to the learning rate.
3. Section 4.4 only shows the performance under different "c"s. More discussions are recommended.

**Summary Of The Paper:**

This paper proposes a novel data augmentation method called context-additive augmentation. This method is easily-implemented, and preserves a target instance without perturbing it. Based on this method, the authors additionally propose a cross-window self-training framework. This framework employs reliability-based cross-window labels to improve consistency among augmented instances. The experiment results show the good performance of the proposed method.

**Summary Of The Review:**

This paper proposes a novel data augmentation method for self-supervised learning. The experiment results show the good performance of the proposed method.

---

> ### Author Response · Authors · 2022-11-16
> **Response to Reviewer Cx6c**
>
> We deeply appreciate the reviewer’s detailed comments on our manuscript.
>
> **Q1. One might don't agree with the rationale in Section 3.3 and the curve $r(p)$ should be. In my opinion, the reliability value of the middle point of the augmented instance is maximum is not proper. In my opinion, the overlap ($m-o$ to $m+o$) should have the largest reliability. If the rationale you proposed holds, assuming that $c < 2o$, the $m-c/2$-th position on $X^\text{left}$ will have much larger reliability than the same position but on $X^\text{right}$. Please explain the intuition of your rationale.**
>
> Let us elaborate on the intuition here. The target instance is a set of timestamps to be pseudo-labeled. Each timestamp $t \in [m-o:m+o)$ can be pseudo-labeled twice from the two augmented instances. Our rationale is based on this setting as follows. The reliability of $m-0.5c$-th position on $X^\text{left}$ should have a larger value than that of $X^\text{right}$ because the pseudo-label at $m-0.5c$-th position on $X^\text{left}$ is predicted with more balance in the number of side data points. We use the same number of data points on the left and right side (i.e., $o+0.5c$) to predict the pseudo-label at $m-0.5c$-th position on $X^\text{right}$. We use a different number of data points on the left ($o-0.5c$) and right ($o+1.5c$) side to predict $\hat{y}^\text{right}_{m-0.5c}$. The balance on each side is important because the prediction would not be biased to either before or after data points.
>
>
> **Q2. It is confused with the augmentation method you proposed. Assuming that the target instance is X[m-o, m+o], the generated instances are X[m-o-c, m+o] and X[m-o, m+o+c]. In my option, this is equal to the case that the target is X[m-o-c, m+o+c] and cut the target into X[m-o-c, m+o] and X[m-o, m+o+c]. So I think the novelty is limited.**
>
> Context-additive augmentation produces two augmented instances from a target instance, one with the left context and another with the right context to make the two augmented instances as *different* as possible. The underlying reason is that forcing consistency between the model outputs from two augmented instances with more difference gives more strong supervision to the model [Wang et al., 2022]. If we use the cutting augmentation you mentioned, the resulting two augmented instances would be too similar, and consistency regularization will give weaker supervision. We emphasized this motivation in Section 3.2.
>
>
> **Q3. In Section 3.2, the authors say that the heuristic "not only maximizes the difference between the two augmented instances but also minimizes the overlap between them". However, I am not sure what is the difference between "maximizing the difference" and "minimizing the overlap". I think the authors just expressed the same meaning in two ways. If there is some difference, please explain it in detail.**
>
> As you said, there is no difference between the two phrases. We erased the sentence for a concise explanation. Thanks for the suggestion.
>
> **Q4. The purpose of Figure 5 is unclear. Since the accuracy has been shown in the tables, and it is difficult to see that CrossMatch converges most quickly. Even if it converges quickly, the speed of convergence is likely related to the learning rate.**
>
> Sorry for your confusion on convergence rate. We used the term convergence rate wrong as there is no difference in convergence speed as you mentioned. Due to correct and sufficient pseudo-labels (which can be shown in pseudo-label F1 in Figure 5), CrossMatch can generate more informative cross-window labels. So, the speed of convergence is faster regardless of a learning rate. We corrected this issue in Section 4.2.

---

> > ### Author Response · Authors · 2022-11-16
> > **Additional Response to Reviewer Cx6c**
> >
> >
> > **Q5. Section 4.4 only shows the performance under different "c"s. More discussions are recommended.**
> >
> > We conducted two more ablation studies on context additive augmentation.
> >
> > > CrossMatch with only-left or only-right augmented instance
> >
> > First, we used either only-left or only-right augmentation in CrossMatch and compared them with the original context-additive augmentation. As the only-left or the only-right augmented instance is similar to the target instance, the effect of consistency regularization becomes weaker, resulting in low classification performance. We confirmed this effect in the following table and reflected in Section 4.4 and Table 6.
> >
> > | Dataset |   Only-left Augmentation  ||    Only-right Augmentation || Context-additive Augmentation||
> > |:-------:|:------------:|:-----------:|:-------------:|:-----------:|:-------------:|:-----------:|
> > |         | TS Accuracy  |    F1@25    |  TS Accuracy  |    F1@25    |  TS Accuracy  |    F1@25    |
> > | HAPT    | $0.72 \pm 0.01$ | $0.65 \pm 0.03$ | $0.73 \pm 0.02$ | $0.67 \pm 0.02$ |  $0.84 \pm 0.01$ |  $0.75 \pm 0.02$ |
> > | mHealth | $0.77 \pm 0.02$ | $0.41 \pm 0.04$ | $0.76 \pm 0.02$ | $0.43 \pm 0.05$ |  $0.85 \pm 0.01$ |  $0.45 \pm 0.02$ |
> >
> > > CrossMatch with the context of a fixed size
> >
> > Second, we used fixed $c$ value instead of random $c$ value (refer to line 3 in Algorithm 1 of Appendix A) and compared their classification performance. Context-additive augmenation with fixed $c$ generates deterministic augmentations because added contexts are always the same if target instance is the same. This hinders the diversity in forcing consistency between two augmented instances resulting in low classification performance. We experimented CrossMatch with a fixed context length $c$ whose value is found as the best in Table 5. The results accord with our intuition as follows.
> >
> >
> > |     Dataset    | CrossMatch with fixed $c$ || CrossMatch with random $c$ ||
> > |:-------:|:------------:|:-----------:|:-------------:|:-----------:|
> > |  | TS Accuracy  |    F1@25    |  TS Accuracy  |    F1@25    |
> > | HAPT    | $0.78 \pm 0.01$| $0.65 \pm 0.02$| $0.84 \pm 0.01$| $0.75 \pm 0.02$|
> > | mHealth | $0.73 \pm 0.02$| $0.22 \pm 0.02$| $0.85 \pm 0.01$| $0.45 \pm 0.02$|

---

### Official Review · Reviewer_gKcX · 2022-10-25

**Confidence:** 3
**Correctness:** 2
**Technical Novelty And Significance:** 3
**Empirical Novelty And Significance:** 3
**Recommendation:** 5

**Clarity, Quality, Novelty And Reproducibility:**

The paper works on an important problem. The technique is okay, but on the heuristic side. But in general it has several unclear parts need to be explained:

1.  What is the target instance for? Is it the instance you would like label, or these are already labeled?
2.  What is the input and output for $f_{theta}$? Please include the used network structure and how many layers you used?
3.  Can you please give an example of your input and output for the overall framework? For example, what is figure 1, what is the time series after your mask? What is the measure for "copy"?  How many labels per instances? Can you give the location and distribution (maybe some heatmap to visualize the original label and the label generated after equation 2, as well as the label after "reliability weighting")? It is really confusing about the problem setting.

Algorithm 1:
For algorithm 1: should line 10 be $Y_u$?


Experiment:
1.  Why table 3 needs to be F1@25 score "over the last 20 iterations"? Why don't we compare with the last iteration directly since there is a proper stop condition?
2. Table 5, it seems like precision was sacrificed while recall is increasing, any comments for it?  Also, the precision and recall do not overlap with the peak with F1@25?
3. How does a fixed c vs a random c value affect the performance?
4. CrossMatch you set tau=0.95, but your pseudo-label per class is trained to 0.99, any reason for that? Is it contributing the performance?
5. Table 4, did you use jitters and scaled jitters when you test the performance with IA or CA?
6.  Also, for the experiment setting, are you concatinating all the instance into a single time series, then perform the pseodo labeling for every dataset? Or you were just doing rolling based on a single instance? I am just confused about the setting.


**Strength And Weaknesses:**

Edit:
Using a segment for context in time series is not really new but contrasting


Strength:
1. The problem is very important since time series are often lacking of labels.
2. The performance is better than the state-of-the-art.

Weakness:
1. unclear problem setting
2. unclear experiment setting
3. the proposed method is very heuristic and not clean.
see below in detailed comments.

**Summary Of The Paper:**

Edit: I would like to thank the authors for explaining their method and updating the scripts. Although they resolved several of my concerns,  I think the proposed method of using context is not very new, but using left and right context has some novelty, but not very high.

As the authors said, the labels are spreading evenly with one class per segment. In addition, the label length is relatively long without switching often. F1 score might not be that sensitive. The problem setting is relatively ideal. I am not sure whether we need this method as maybe some simpler solution would work in this case.

Thus, my score will remain the same.


---------------------------------------------------------------------------------------------------------------------------------------------------------
Original review:

This paper proposed a new framework for self-training for generating pseudo labels for a long time series. The proposed framework propose some training strategy on the few label cases and then use post-processing by utilizing the left and the right context to reweight the labels.



**Summary Of The Review:**

I will consider raising my evaluation if my concerns are addressed by the authors. I think the problem setting, the overall model, and the experiment setting needs more explanation.

---

> ### Author Response · Authors · 2022-11-16
> **Response to Reviewer gKcX**
>
> We deeply appreciate the reviewer’s constructive comments and positive feedback on our
> manuscript.
>
> **Q1. What is the target instance for? Is it the instance you would like label, or these are already labeled?**
>
> A target instance is an instance we would like to do pseudo-labeling, where its only few timestamps might have been labeled. We clarified this meaning in Section 1 and Figure 1(a).
>
>
> **Q2. What is the input and output for f_theta? Please include the used network structure and how many layers you used?**
>
> > What is the input and output for f_theta?
>
> As mentioned in Section 4.1, we used a MS-TCN as $f_\theta$. MS-TCN is the abbreviation of a multi-stage temporal convolutional network [Farha et al., 2019]. It can classify each data point in a segment instance $X$, generating softmax probability distribution at each timestamp.
>
> > Please include the used network structure and how many layers you used?
>
> MS-TCN has four stages and each stage is composed of eleven dilated convolution layers and a single softmax output layer. The first stage takes a subsequence of the whole time series and outputs softmax probabilities at each timestamp. After the first stage, every stage is fed with softmax probabilities and then outputs another softmax probabilities.
>
> We included the model description in Appendix B.
>
> **Q3. Can you please give an example of your input and output for the overall framework? For example, what is figure 1, what is the time series after your mask? What is the measure for "copy"? How many labels per instance? Can you give the location and distribution (maybe some heatmap to visualize the original label and the label generated after equation 2, as well as the label after "reliability weighting")? It is really confusing about the problem setting.**
>
> > what is the figure 1?
>
> We updated Figure 1 for better understanding of the overall procedure of CrossMatch. Figure 1(a) illustrates context-additive augmentation from a target instance, generating pairs of augmented instances as a training batch. Figure 1(b) shows the self-training procedure of a pair of augmented instances.
>
> > what is the time series after your mask?
>
> Specifically in Figure 1(b), CrossMatch pseudo-labels the timestamps only in the target instance, not in the added context. So, each model output of an augmented instance is masked to consider only the outputs for the timestamps in the target instance.
>
> > what is the measure for copy?
>
> Sorry for the confusion about copy process. From the pair of masked model outputs, CrossMatch generates  cross-window soft labels and share them with the two model outputs $f_\theta(X^\text{left})$ and $f_\theta(X^\text{right})$ for cross-entropy loss computation. We indicated this sharing process as copy in the previous version but injected it into the loss formula $\ell_u$ in the current version for clarity.
>
> > How many labels per instance?
>
> Each timestamp in a target instance can have a cross-window soft label if at least one of the augmented instances has a confident pseudo-label (Equation (2) in Section 3.1). So, the maximum number of labels per instance is the number of unlabeled timestamps in a target instance.
>
> > Can you give the location and distribution (maybe some heatmap to visualize the original label and the label generated after equation 2, as well as the label after "reliability weighting")?
>
> In Figure 1, we visualized a target instance with original labels, a window randomly sampled from sparsely-labeled time series. CrossMatch outputs cross-window soft labels after reliability-weighted mixing from two softmax probabilities from augmented instances. The shade has two colors at a timestamp if  pseudo-labels are generated in both augmented instances; otherwise it has a single color. If no pseudo-labels exist, then there is no cross-window label, illustrated as the white shade. We tried to design effective context-additive augmentation and generate reliable labels for self-training in sparsely-labeled time series.
>
> We added a more specific problem setting in Section 1 and Section 3.1 and summarized the overall procedure in Figure 1 and Section 3.4 for better understanding. Please let us know if you need more clarification with the revision.
>
> **Q4. For algorithm 1: should line 10 be Y_u?**
>
> Thank you for the correction. We updated it accordingly.
>
>
>
> [Farha and Gall, 2019] MS-TCN: Multi-stage temporal convolutional network for action segmentation. In CVPR, 2019.

---

> > ### Author Response · Authors · 2022-11-16
> > **Additional response to Reviewer gKcX**
> >
> >
> > **Q5. Why table 3 needs to be F1@25 score "over the last 20 iterations"? Why don't we compare with the last iteration directly since there is a proper stop condition?**
> >
> > Due to the unstability in pseudo-labeling in semi-supervised learning, it is not practical to report only the last evaluation result. To get more robust evaluation results, following the previous work [Berthelot et al., 2019], we first trained all algorithms with the same number of total iterations which is enough for all algorithms to be converged, and then reported the average evaluation results of the last 20 iterations.
> >
> >
> >
> > **Q6. Table 5, it seems like precision was sacrificed while recall is increasing, any comments for it? Also, the precision and recall do not overlap with the peak with F1@25?**
> >
> > > Trade-off between precision and recall of pseudo labels.
> >
> > In Figure 7 (i.e., Figure 5 in the original draft that you may have mistakenly referred it to as Table 5 in your question), pseudo-label precision is high and recall is low at the early stage of self-training iterations, but the precision becomes lower and the recall becomes higher after sufficient iterations. This is because only a few easy data points are pseudo-labeled in the early stage, while many hard data points are pseudo-labeled later.
> >
> > > Relationship between pseudo labeling performance (precisian and recall) and classification performance (F1@25 and TS Accuracy).
> >
> > The peak of the precision or recall may not overlap the peak of classification performance, but the F1 score computed from the precision and recall shows some overlap. This overlap is shown in Figure 5 in the revised draft.
> >
> >
> > **Q7. How does a fixed c vs a random c value affect the performance?**
> >
> > Context-additive augmenation with fixed $c$ generates deterministic augmentations because added contexts are always the same if the target instance is the same. This hinders the diversity in forcing consistency between two augmented instances, resulting in low classification performance. We experimented CrossMatch with a fixed context length $c$ whose value is found as the best in Table 5 where the ablation study for the optimal context length is done. The result confirms our expectation, as below.
> >
> >
> > |     Dataset    | CrossMatch with fixed $c$ || CrossMatch with random $c$ ||
> > |:-------:|:------------:|:-----------:|:-------------:|:-----------:|
> > |  | TS Accuracy  |    F1@25    |  TS Accuracy  |    F1@25    |
> > | HAPT    | $0.78 \pm 0.01$| $0.65 \pm 0.02$| $0.84 \pm 0.01$| $0.75 \pm 0.02$|
> > | mHealth | $0.73 \pm 0.02$| $0.22 \pm 0.02$| $0.85 \pm 0.01$| $0.45 \pm 0.02$|
> >
> > We made a new table, Table 6, in the manuscript for this result.
> >
> > **Q8. CrossMatch you set tau=0.95, but your pseudo-label per class is trained to 0.99, any reason for that? Is it contributing the performance?**
> >
> >
> > > The difference between the threshold for confidence $\tau$ vs. the threshold for entropy of class size.
> >
> > The two hyperparameters have different roles. The confidence threshold ($\tau$) determines which unlabeled timestamps should be pseudo-labeled, while the entropy threshold ($0.99$) decides whether or not to initiate pseudo-labeling during training. To compute the entropy, we first count the number of each class in pseudo-labels and compute the ratio of each class. Using the ratio value, we can compute the entropy of class size in pseudo-labels.
> >
> >
> > > Contribution of the confidence threshold $\tau$ and the entropy of class size.
> >
> > If we lower the confidence thresholds, more unconfident predictions are used as pseudo-labels, increasing recall but decreasing precision. If we lower the entropy threshold, pseudo-labeling starts earlier even pseudo-labels are not balanced in their class distribution. In other words, The confidence threshold controls the tradeoff between pseudo-label quality and quantity. The entropy threshold controls the tradeoff between early training and class imbalance in pseudo-labels.
> >
> >
> > **Q9. Table 4, did you use jitters and scaled jitters when you test the performance with IA or CA?**
> > Yes, we used jittering as a weak augmentation and scaled jittering as a strong augmentation for IA. Note that original FixMatch, FlexMatch, and PropReg exploit IA for consistency regularization between the outputs from the weak and strong augmentations.
> >
> >
> > [Berthelot et al., 2019]. Mixmatch: A holistic approach to semi-supervised learning. In NeurIPS, 2019.

---

> > > ### Author Response · Authors · 2022-11-16
> > > **Additional Response to Reviewer gKcX**
> > >
> > >
> > > **Q10. Also, for the experiment setting, are you concatenating all the instance into a single time series, then perform the pseudo labeling for every dataset? Or you were just doing rolling based on a single instance? I am just confused about the setting.**
> > >
> > > Thank you for pointing out our mistake in writing. We did not concatenate all the instances. For each fully-labeled dataset, we randomly sample the equal number of timestamps for each class and drop the labels at the rest of timestamps to generate *sparsely-labeled* time series. The sampled timestamps become a labeled timestamp set $\mathcal{T}_L$ given the ratio of labeled timestamps $z=|\mathcal{T}_L|/|\mathcal{T}|$. For self-training, we randomly sampled a target instance from the sparsely-labeled time series and pseudo-label the target instance as shown in Figure 1. We corrected our manuscript accordingly in Section 4.1.

---

> > > > ### Comment · Reviewer_gKcX · 2022-11-17
> > > > **Response to the authors**
> > > >
> > > > I appreciate the detailed response from the authors. It helps on understanding the setting of the problem, so I can better evaluate the paper now. There are several concerns from me:
> > > >
> > > > 1. The paper is working on pseudo-labeling for time series points not aiming for segments, I am not sure whether this is meaningful enough because in high-resolution data, the information is actually on the segment level. I am not sure the evaluation is meaningful enough.
> > > >
> > > > 2. The ratio of 0.1% label is kind of misleading. In sensor data, especially in activity, many labels in nearby intervals are essentially the same and can be easy to propagate.  What matters how often would a class switch to the next class? For example, the activity of running could last for thousands of points before switching to the next activity. There is no description in the dataset about it. There is also lacking of information about the distribution of labels in the original dataset. Are the labels spreading evenly in the data or only existing in a very small time span?
> > > >
> > > > 3. In section 4.1, “We averaged F1 score over the target instances in a batch. When there is no pseudo-label in a batch, both of precision and recall become 0, which also makes F1 score as 0.” Can you elaborate more? It seems like you are evaluating on only the neighborhood of the labeled data and large amount of data were ignored although the dataset is fully labeled? Shouldn’t the entire data be used?
> > > >
> > > > 4. One other concern is on Table 2, I work on time series but I don’t work on pseudo labelling personally, the iteration number is unnecessarily large ranging from 25000 to 50000. Typically a classification model will need a few hundred iteration to converge. Even FixMatch (Sohn et al) on image data, the experiment stated that they only trained 300 epochs with 5 epochs for warm up. Why the proposed model needs these huge epochs? Also, it makes the comparison sort of unconvincing and the model impractical to use for real-world applications.
> > > >
> > > > 5. Minor question, how large is the segment in segment F1 evaluation? Are the Y segments overlapping?

---

> > > > > ### Author Response · Authors · 2022-11-18
> > > > > **Second Response to Reviewer gKcX**
> > > > >
> > > > > **Q1. The paper is working on pseudo-labeling for time series points not aiming for segments, I am not sure whether this is meaningful enough because in high-resolution data, the information is actually on the segment level. I am not sure the evaluation is meaningful enough.**
> > > > >
> > > > > Segment-level information is not always available in various applications such as fall detection and action segmentation. A classifier should predict each timestamp for an immediate decision and fine-granular classification. So we define that a target instance can consist of homogeneous labels with a single class or heterogeneous labels with multiple classes. As a result, we evaluate the classifier in both timestamp and segment levels, as described in Section 4.1 and Appendix C.
> > > > >
> > > > >
> > > > > **Q2. The ratio of 0.1% label is kind of misleading. In sensor data, especially in activity, many labels in nearby intervals are essentially the same and can be easy to propagate. What matters how often would a class switch to the next class? For example, the activity of running could last for thousands of points before switching to the next activity. There is no description in the dataset about it. There is also lacking of information about the distribution of labels in the original dataset. Are the labels spreading evenly in the data or only existing in a very small time span?**
> > > > >
> > > > > > In sensor data, especially in activity, many labels in nearby intervals are essentially the same and can be easy to propagate. What matters how often would a class switch to the next class? There is no description in the dataset about it.
> > > > >
> > > > > In addition to the answers in Q1, class segments can last for different durations as a single class, which makes it hard to propagate a small amount of sparse labels accurately. *Assuming* that we know the ground-truth segments, the proportion of the segments containing a labeled timestamp can be calculated for each dataset (HAPT: 30/451, mHealth: 60/117, Opportunity: 34/1728). We can also get the average length of a class segment (i.e., the average number of data points before switching to the next class), as reported in Table 2 (the column name is abbreviated as Length). Note that CrossMatch does not have any access to the ground truth about the segment-level information mentioned above. We revised Section 4.1 to mention this setting.
> > > > >
> > > > >
> > > > > > There is also lacking of information about the distribution of labels in the original dataset. Are the labels spreading evenly in the data or only existing in a very small time span?
> > > > >
> > > > > The labels are spread *evenly* in time series (due to random sampling for creating a sparsely-labeled time series) while maintaining the same number of labels for each class. We described this setting in Section 4.1 of the revised manuscript.
> > > > >
> > > > >
> > > > > **Q3. In section 4.1, “We averaged F1 score over the target instances in a batch. When there is no pseudo-label in a batch, both of precision and recall become 0, which also makes F1 score as 0.” Can you elaborate more? It seems like you are evaluating on only the neighborhood of the labeled data and large amount of data were ignored although the dataset is fully labeled? Shouldn’t the entire data be used?**
> > > > >
> > > > > At each iteration, we measure precision and recall from each timestamp of target instances in a batch, as follows:
> > > > > $$\text{Precision}=\frac{\text{the number of correct PLs}}{\text{the number of PLs}} \quad\quad\quad\text{Recall}=\frac{\text{the number of correct PLs}}{\text{the number of timestamps}}$$
> > > > > For *every* timestamp, we check the existence of a pseudo-label and the class of the true label to see if the class of a pseudo-label is correct. When there is no pseudo-label at any timestamp, the denominator of precision becomes 0 and the precision can diverge. So, we define the precision with no pseudo-label as 0. After a few iterations, the *entire* data is expected to be used by pseudo-labeling the sampled target instances. We added this explanation in Section 4.1 and Appendix C. Again, the entire data is used for the evaluation of pseudo-labeling.
> > > > >
> > > > > [Moltisanti et al., 2019] Action recognition from single timestamp supervision in untrimmed videos, In CVPR. 2019.
> > > > >
> > > > > [Ma et al., 2020] SF-net: Single-frame supervision for temporal action localization, In ECCV, 2020.
> > > > >
> > > > > [Li et al., 2021] Temporal action segmentation from timestamp supervision. In CVPR. 2021.

---

> > > > > > ### Author Response · Authors · 2022-11-18
> > > > > > **Additional Second Response to Reviewer gKcX**
> > > > > >
> > > > > >
> > > > > >
> > > > > > **Q4. One other concern is on Table 2, I work on time series but I don’t work on pseudo labelling personally, the iteration number is unnecessarily large ranging from 25000 to 50000. Typically a classification model will need a few hundred iteration to converge. Even FixMatch (Sohn et al) on image data, the experiment stated that they only trained 300 epochs with 5 epochs for warm up. Why the proposed model needs these huge epochs? Also, it makes the comparison sort of unconvincing and the model impractical to use for real-world applications.**
> > > > > >
> > > > > > Iteration is a smaller unit for the number of repetitions compared with Epoch (i.e., $\text{1 Epoch}=\frac{|\mathcal{T}|}{2o*B}\,\text{Iteration}$). We ran every dataset for about 25--50 epochs, which takes about 2 hours at most, using Intel(R) Xeon(R) Gold 6226R CPU @ 2.90GHz and NVIDIA RTX 3090.
> > > > > >
> > > > > >
> > > > > > **Q5. Minor question, how large is the segment in segment F1 evaluation? Are the Y segments overlapping?**
> > > > > >
> > > > > > As answered in Q2, the average length of true segments is in Table 2. Each timestamp can have only one class so the segments do not overlap.

---

> > > > > > > ### Author Response · Authors · 2022-12-05
> > > > > > > **Response to edited review**
> > > > > > >
> > > > > > > Sorry for the late response to your edited review because there was no email notification for editting. We want to emphasize that classifying long time series is not a simple problem when we do not know the boundaries of class segments. When we train a sequential classifier with *full timestamp label*, over-segmentation(i.e., too many switchings) errors often occur, which leads to low segmental F1 score [Farha et al., 2019, Wang et al., 2020]. This problems would be more severe when a time series has sparse labels.
> > > > > > >
> > > > > > > In our experiment setting, a time series is composed of multiple segments whose lengths are all different. A segment is composed of data points which are indexed with timestamps. The data points in a segment have the same class label. For sparsely-labeled time series, we select multiple chunks of consecutive timestamps in each segment randomly, preserving the same number of labeled data points for each class. As a result, every class has the same number of labels but the location is unknown. For example, the labels for a certain class may reside in a single segment although the time series have multiple segments with the class. Evenly distribution of the labels does not mean that at least one data point is labeled for each segment.
> > > > > > >
> > > > > > > We deal with challenging problem that even fully-supervised classifier cannot solve easily and tried our best to make realistic sparsely-labeled time series.
> > > > > > >
> > > > > > >
> > > > > > > [Farha et al., 2019]. MS-TCN: Multi-stage temporal convolutional network for action segmentation. In CVPR. 2019.
> > > > > > >
> > > > > > > [Wang et al., 2020]. Boundary-aware cascade networks for temporal action segmentation. In ECCV, 2020.

---

### Official Review · Reviewer_4aZf · 2022-11-01

**Confidence:** 4
**Correctness:** 4
**Technical Novelty And Significance:** 3
**Empirical Novelty And Significance:** 2
**Recommendation:** 6

**Clarity, Quality, Novelty And Reproducibility:**

Ideas and suggestions on how to improve the clarity quality and the presentation of this work are mentioned above in the weaknesses. The code and the data will be available online for reproducibility. The novelty of the paper is incremental as discusses above.

**Details Of Ethics Concerns:**

No ethical concerns. The authors have added a paragraph commenting on ethical concerns regarding the anonymization of the users in the datasets.

**Strength And Weaknesses:**

Strengths:
- The paper is well-written and easy to follow. The authors have done a good job in the writing and the structure of the paper.
- The problem is interesting to the research community.
- The authors have cover the related works and show what are the current challenges and how the proposed method will help on those and increase performance.
- The authors have conducted an extensive experimental setup and the results are very promising as they show that in most cases the proposed model outperforms the comparison methods.
- The authors have included analyses about the augmentations types that have been selected and the varying context length for the augmented instances.
- The authors will release the code.

Weaknesses:
- The novelty of the paper is incremental as the authors rely on existing methodology, however they do add extra analyses and they do have their own contributions.
- It would be interesting to discuss examples of the additive augmentations that have been successful from the datasets in the experimental setup.
- The whole idea of either left or a right augmentation needs more motivation. Why do the authors propose only one of the two? Because in this concept the time is changing, how do only left (previous TS segments) perform? How does the performance get affected by only right additive augmentations? What does that mean for the TS segments on the right (future)? Have the authors experiment with combinations of the augmentations?
- The idea of the notation and equations etc to be calculated based on the half and the middle timestamp of the time-series segment is a bit confusing, wouldn’t it be easier to follow if the paper was written with 1/2*{length of X’} for example?
- In the definitions of the evaluation metrics, what is the relation of the segmental accuracy and F1-Score to Jaccard similarity? This particular paragraph needs a better writing, as it is not clear how the segmental scores are being calculated, which is important for the understanding of the results.
- In Table 3, TS Accuracy in FixMatch, FlexMatch and PropReg should all be in bold, as they are all the same.
- The results in the tables of the experimental setup, have small differences between the proposed and the comparison methods. Have the authors checked for statistical significance in those results?
- It is not clear what the iteration axis refers to in Figure 5.

**Summary Of The Paper:**

In this paper, a method for time series self training is being proposed. The model is named CrossMatch and it utilizes context-additive augmentation, in other words, it adds a context instance to the left and the right part of the time-series to generate two augmented views with different contexts. Then, the authors design a reliability function for more reliable pseudo-labels, and they mix the pseudo-labels into a single cross-window label, which will be matched against two softmax probabilities from both of augmentations. The CrossMatch model is test against three other state of the art models (FixMatch, FlexMatcha and PropReg) in three publicly available datasets (HAPT, mHealth and Opportunity). The results show that in most cases the proposed model outperforms the rest in segmented Accuracy and segmented F1 Score. In the experimental setup, the authors also show results for the inter- and inner-instance augmentations analysis and an analysis of varying context length for the augmented instances.

**Summary Of The Review:**

Overall, I enjoyed reading this paper. It is easy to follow, well-written and well-structured. The authors do a good job providing the motivation, the challenges and the related works. A few suggestions for improvements can be found in this review. The results show that the proposed model outperforms the rest of the comparison methods in three real world datasets..

---

> ### Author Response · Authors · 2022-11-16
> **Response to Reviewer 4aZf**
>
> We deeply appreciate the reviewer's detailed comments on our paper.
>
> **Q1. The whole idea of either left or a right augmentation needs more motivation. Why do the authors propose only one of the two? Because in this concept the time is changing, how do only the left (previous TS segments) perform? How does the performance get affected by only right additive augmentations? What does that mean for the TS segments on the right (future)? Have the authors experimented with combinations of the augmentations?**
>
> > The whole idea of either left or a right augmentation needs more motivation. Why do the authors propose only one of the two?
>
> The reason is to force a consistency between the model outputs from two augmented instances with a *large difference*, giving more strong supervision to the model [Wang et al., 2022]. Therefore, context-additive augmentation produces two augmented instances, which are *as different as possible*, from a target instance: one with the left context and another with the right context.
> We emphasized this motivation in Section 3.2.
>
> >What does that mean for the TS segments on the right (future)?
>
> The time series segment on the *right* is collected *after* a target instance, whereas the time series segment on the *left* is collected *before* a target instance. CrossMatch samples a target instance randomly from already collected time series, so the left and right contexts are mostly available. Of course, the start and the end of the time series are not sampled to reserve the left and right context.
>
> > Because in this concept the time is changing, how do only the left (previous TS segments) perform? How does the performance get affected by only right additive augmentations?  Have the authors experimented with combinations of the augmentations?
>
> As the only-left or only-right augmented instance is similar to the target instance, the effect of consistency regularization becomes weaker, resulting in low classification performance. We confirmed this effect in the following table and reflected in Section 4.4 and Table 6.
>
> | Dataset |   Only-left Augmentation  ||    Only-right Augmentation || Context-additive Augmentation||
> |:-------:|:------------:|:-----------:|:-------------:|:-----------:|:-------------:|:-----------:|
> |         | TS Accuracy  |    F1@25    |  TS Accuracy  |    F1@25    |  TS Accuracy  |    F1@25    |
> | HAPT    | $0.72 \pm 0.01$ | $0.65 \pm 0.03$ | $0.73 \pm 0.02$ | $0.67 \pm 0.02$ |  $0.84 \pm 0.01$ |  $0.75 \pm 0.02$ |
> | mHealth | $0.77 \pm 0.02$ | $0.41 \pm 0.04$ | $0.76 \pm 0.02$ | $0.43 \pm 0.05$ |  $0.85 \pm 0.01$ |  $0.45 \pm 0.02$ |
>
>
> **Q2. It would be interesting to discuss examples of the additive augmentations that have been successful from the datasets in the experimental setup.**
>
> A motion-sensing time-series dataset, mHealth, contains a action segment labeled with class`Jumping` ($t=1,\ldots,12$) composed of several sub-action features such as Lifting ($t=1,2,3$), Midair($t=4,\ldots,9$), and Landing ($t=10,11,12$). From a target instance $\dot{X}$ that only contains Midair ($t=4,\ldots,9$), context-additive augmentation generates two augmented instances $X^\text{left}$ that contains Lifting + Midair ($t=1,\ldots,9$) and $X^{\text{right}}$ that contains Midair + Landing ($t=4,\ldots,12$). If we have a pseudo-label with class `Jumping` at a timestamp in $\dot{X}$ from two model outputs $f_\theta(X^{\text{left}})$ and $f_\theta(X^{\text{left}})$, CrossMatch shares the pseudo-label as a prediction target $f_\theta(X^{\text{left}})$ and $f_\theta(X^{\text{left}})$. From the classification loss after sharing, the model can learn the correlation between `Jumping` with two different contexts, Lifting and Landing. As a result, an augmented instance mostly covered with Lifting or Landing would have a pseudo-label of `Jumping` with more probability, leading to more effective self-training.
>
> We will add more ample examples in our manuscript soon.
>
>
> **Q3. The idea of the notation and equations etc to be calculated based on the half and the middle timestamp of the time-series segment is a bit confusing, wouldn't it be easier to follow if the paper was written with 1/2\*{length of X’} for example?**
>
> Thanks for your suggestion on the notation. We chose to use half-lengths and the middle timestamp for better readability in math formulas and figures. We will look into it more deeply.
>
> [Wang et al., 2022] On the importance of asymmetry for siamese representation learning. In CVPR, 2022.

---

> > ### Author Response · Authors · 2022-11-16
> > **Additional Response to Reviewer 4aZf**
> >
> >
> > **Q4. In the definitions of the evaluation metrics, what is the relation of the segmental accuracy and F1-Score to Jaccard similarity? This particular paragraph needs better writing, as it is not clear how the segmental scores are being calculated, which is important for the understanding of the results.**
> >
> > Sorry to make you confused about our evaluation metrics. Segmental F1 score is the same as segmental accuracy, and Jaccard similarity is the same as the intersection over union (IoU). This is a popularly used metric to measure segment prediction performance for judging whether a classifier outputs *correct and coherent* labels for consecutive timestamps. [Kumar et al., Li et al., Lea et al.].
> >
> > We made the explanation more concise and updated Section 4.1 and Appendix C.
> >
> > **Q5. In Table 3, TS Accuracy in FixMatch, FlexMatch and PropReg for HAPT (10x) should all be in bold, as they are all the same.**
> >
> > Thanks for pointing out the error. We corrected Table 3.
> >
> >
> > **Q6. The results in the tables of the experimental setup, have small differences between the proposed and the comparison methods. Have the authors checked for statistical significance in those results?**
> >
> > In independent (unpaired) t-test, we verified that the results of CrossMatch are significantly higher than the results of compared algorithms with p-value < 0.05, except for HAPT (10x). We updated Section 4.2 to include this statistical analysis result.
> >
> >
> > **Q7. It is not clear what the iteration axis refers to in Figure 5.**
> >
> > An interation indicates a single step of sampling/augmenting instances, computing losses, and updating model parameters, as detailed in Algorithm 1 of Appendix A. We clarified this in Figure 5.
> >
> >
> >
> > [Li et al., 2021] Temporal action segmentation from timestamp supervision. In CVPR, 2021.
> >
> > [Kumar et al., 2022] Unsupervised action segmentation by joint representation learning and online clustering. In CVPR, 2022.
> >
> > [Lea et al. 2017] Temporal convolutional networks for action segmentation and detection. In CVPR, 2017.

---

### Author Response · Authors · 2022-11-16
**General response to all reviewers**

Dear Reviewers, we are grateful to have your sincere comments on our paper. We have prepared our rebuttal (in sections below) and revised the manuscript carefully according to the comments. The revised parts are highlighted in color and labeled with the reviewer ID (R1: 4aZf, R2: gKcX, R3: Cx6c, and R4: Rri3).

---

### Author Response · Authors · 2022-12-05
**Looking for further discussion**

Dear reviewers, would you advise if we, the authors, have fully addressed all of your comments in our rebuttal and revision?  Please let us provide below a summary of our rebuttal and revision.

* Reviewer [4aZf](https://openreview.net/forum?id=whsWWPAUkwR&noteId=3CMlOApS1V): To clarify the motivation in context-additive augmentation, we emphasized that the model outputs from two augmented instances with a *large difference* gives more strong supervision. An additional experiment that verifies this motiviation was also done. We made the description about the evluation metrics more concise and checked statistical significance of the experiment results. We also corrected the mistake in numerics in Table 3.

* Reviewer [gKcX](https://openreview.net/forum?id=whsWWPAUkwR&noteId=E4-yVYo0AL): Thanks for your reply again. After your second review, we explained the meaning of pseudo-labeling for time series points, using the applications such as fall detection and action segmentation. We clarified the experiment setting on sparsely-labeled time series where a class segment can last for different durations. We also elaborated on the explanations of the pseudo-label F1 score, the number of training iterations, and the length of a segment to address your concerns.

* Reviewer [Cx6c](https://openreview.net/forum?id=whsWWPAUkwR&noteId=QRQxpxCXwo): We explained our intuition in the reliability function which shows the highest value at the center of an augmented instance. We described the advantage of context-additive augmentation in consistency regularization with a citation. We added two additional experiments to discuss more the performance of CrossMatch under different context lengths $c$. We corrected some sentences and explanations that you mentioned.

* Reviewer [Rri3](https://openreview.net/forum?id=whsWWPAUkwR&noteId=DRkPHKLIl1): In response to your review of the introduction, we added the problem setting and described several terms in detail. We also explained the challenge and motivation in terms of consistency regularization for clarification. We corrected our description of the dataset and explained the assumptions in sparsely-labeled time series. We made some justifications about not comparing other label propagation methods and the fluctuation in the number of pseudo-labels.

---

### Decision · Program_Chairs · 2023-01-20

**Decision:**

Reject

**Justification For Why Not Higher Score:**

Incremental contribution, limited comparison to the relevant alternatives.

**Justification For Why Not Lower Score:**

n/a

**Metareview: Summary, Strengths And Weaknesses:**

This interesting paper has been assessed by four knowledgeable reviewers two of whom rated it as marginal reject and two as marginal accept. The key complaints include incremental novelty some issues with clarity and fairness of comparison versus relevant alternatives. The authors have actively engaged in extensive discussions with some reviewers during the rebuttal period, but these discussions and provided clarifications did not lead to improved ratings. This work in its current form needs improvements, it is not too far from being publishable, so the authors should be encouraged to push forward, but at its current state it does not meet the acceptance criteria for ICLR.